# Genetics and evidence for balancing selection of a sex-linked colour polymorphism in a songbird

Kang-Wook Kim [1,8], Benjamin C. Jackson [1,8], Hanyuan Zhang[1], David P.L. Toews [2,3,4], Scott A. Taylor [2,3,5], Emma I. Greig[2], Irby J. Lovette [2,3], Mengning M. Liu [6], Angus Davison [6], Simon C. Griffith [7], Kai Zeng [1] & Terry Burke [1]

Colour polymorphisms play a key role in sexual selection and speciation, yet the mechanisms that generate and maintain them are not fully understood. Here, we use genomic and transcriptomic tools to identify the precise genetic architecture and evolutionary history of a sex-linked colour polymorphism in the Gouldian finch *Erythrura gouldiae* that is also accompanied by remarkable differences in behaviour and physiology. We find that differences in colour are associated with an ~72-kbp region of the Z chromosome in a putative regulatory region for *follistatin*, an antagonist of the *TGF-β* superfamily genes. The region is highly differentiated between morphs, unlike the rest of the genome, yet we find no evidence that an inversion is involved in maintaining the distinct haplotypes. Coalescent simulations confirm that there is elevated nucleotide diversity and an excess of intermediate frequency alleles at this locus. We conclude that this pleiotropic colour polymorphism is most probably maintained by balancing selection.

[1] Department of Animal and Plant Sciences, University of Sheffield, Sheffield S10 2TN, UK. [2] Fuller Evolutionary Biology Program, Cornell Lab of Ornithology, Cornell University, 159 Sapsucker Woods Road, Ithaca, NY 14850, USA. [3] Department of Ecology and Evolutionary Biology, Cornell University, Corson Hall, Ithaca, NY 14853, USA. [4] Department of Biology, The Pennsylvania State University, University Park, PA 16802, USA. [5] Department of Ecology and Evolutionary Biology, University of Colorado Boulder, Boulder, CO 80309, USA. [6] School of Biology, University of Nottingham, University Park, NG7 2RD Nottingham, UK. [7] Department of Biological Sciences, Macquarie University, Sydney, NSW 2109, Australia. [8] These authors contributed equally: Kang-Wook Kim, Benjamin C. Jackson. Correspondence and requests for materials should be addressed to K.-W.K. (email: k.kim@sheffield.ac.uk) or to K.Z. (email: k.zeng@sheffield.ac.uk) or to T.B. (email: t.a.burke@sheffield.ac.uk)

Colour-polymorphic species are observed across diverse taxa, but are relatively rare and younger than monomorphic species[1,2]. This is probably because any difference in fitness between alternative morphs should result in either the fixation of one of the causative alleles under natural or sexual selection, or lead to sympatric speciation under disruptive selection—potentially increasing the rate of speciation and promoting the transition from polymorphism to monomorphism[2–4]. Even if there are no fitness differences, genetic drift is expected to remove alternative morphs from a small population[3]. Therefore, the evolutionary processes that maintain colour polymorphisms in some species over long periods are not well understood. This is partly because the genetic basis of pigmentation, especially for non-melanic pigments such as carotenoids and psittacofulvins, has only recently been elucidated[5–8].

Gouldian finch (*Erythrura gouldiae*) head plumage is a classic example of a Mendelian colour polymorphism in which the morphs co-occur in sympatry. The striking head colour morphs of the Gouldian finch reflect variation in both carotenoid and melanin pigmentation[9–11]; a pigmentation locus on the Z chromosome (hereafter the *Red* locus, with a recessive $Z^r$ allele and a less frequent dominant $Z^R$ allele) produces melanin-based black ($Z^rZ^r$ in male, $Z^r$ in female) or carotenoid-based red ($Z^RZ^R$, $Z^RZ^r$ in male, $Z^R$ in female) head morphs in both sexes, as well as differences in feather structure[9,10]. Rarely in the wild (<1% of birds), an epistatic interaction of the *Red* locus with an autosomal locus produces a yellow morph[11,12].

With regards to the black and red morphs, there are remarkable inter-morph differences in stress hormone levels, personalities, social dominance, sperm-length plasticity and sex allocation associated with the colour polymorphism (see Supplementary Table 1 for summary and references). Moreover, pre- and postzygotic incompatibilities have been documented between these morphs in a captive population (Supplementary Table 1), but the same phenomena have not been fully investigated in wild populations[13].

Despite these differences, red and black morphs coexist in natural populations at stable phenotypic frequencies[12], suggesting that there are mechanisms maintaining this multi-functional colour polymorphism in nature. Given that multiple phenotypes are associated with head colour, one hypothesis is that the *Red* locus is a supergene, defined as a tight cluster of genes acting as a mechanical unit in particular allelic combinations[14], which is often facilitated by a chromosomal inversion, as recently demonstrated in several avian species[15–18]. Alternatively, the *Red* locus might be a single pleiotropic locus that can regulate multiple genes[19]. We undertook a detailed analysis of the region associated with the colour polymorphism using genomic and transcriptomic tools, and identify the genomic location of the *Red* locus as a putative regulatory region for *follistatin* on the Z chromosome. In addition, by using coalescent simulations, we find evidence that balancing selection is the most probable evolutionary mechanism behind the maintenance of the colour polymorphism in the Gouldian finch.

## Results and Discussion

**Identification of the *Red* locus**. By using high-depth/low SNP density RADSeq[20] and low-depth/high SNP density whole-genome sequencing (see Methods), we identified a region on the Z chromosome strongly associated with red–black head colouration (1000 permutations, $P < 0.05$, Fig. 1a, Supplementary Figs. 1 and 2) within the candidate interval identified in a previous linkage analysis[21]. The genotypes of the wild birds ($n_{male} = 89$, $n_{female} = 72$) from an allele-specific PCR in this region showed a near-perfect association with head colour, with one exception,

in a female (Fisher's exact test, male: $P < 10^{-15}$; female: $P = 7.52 \times 10^{-10}$, Fig. 1b, Supplementary Fig. 2). The frequencies of the black-linked allele in this population were similar in females (0.88) and males (0.85) (Fisher's exact test, $P = 0.69$). The genotypic frequencies in males did not depart from Hardy–Weinberg Equilibrium ($P_{permutation} = 0.24$, Supplementary Table 3). The low frequency of homozygous red males is not significantly different from expectation for a sex-linked locus under random mating (Supplementary Table 3) and a prediction based on the phenotypic frequency[9].

**High LD but no genomic rearrangements around the *Red* locus**. Next, we examined if the multi-functional polymorphism is caused by a supergene. Previously-described supergenes have large genomic blocks of elevated linkage disequilibrium (LD) that harbour multiple genes. We measured the extent of LD around the *Red* locus, defined as the region with the most significant association with the colour polymorphism and its flanking regions (Fig. 1c), using Sanger sequencing of multiple loci. Intriguingly, the LD block significantly associated with the colour polymorphism was limited to a single intergenic region (where no known protein-coding genes are annotated in zebra finch) that spans ~72 kilobase pairs (kbp) and lies between the coding regions of the genes *molybdenum cofactor synthesis 2* (*MOCS2*) and *follistatin* (*FST*) (Fig. 1c). In this region, the density of fixed differences between the $Z^R$ and $Z^r$ alleles was high (6.8–27.8 per kbp; Supplementary Table 5). This region, which is also supported by the analyses of diversity, divergence and simulations (below), is hereafter considered to comprise the *Red* locus.

Centromeres and genomic rearrangements, particularly chromosomal inversions, may reduce recombination rates and generate significant LD blocks between haplotypes[22]. However, our previous work showed that the *Red* locus is not located near the centromere or in a large genomic rearrangement between morphs[21]. We confirmed this finding with a fine-scale test for inversion breakpoints using long-range PCR: our genomic assembly across the *Red* locus for each genotype contained contiguous homologous sequence, consistent with the absence of an inversion differentiating the $Z^R$ and $Z^r$ alleles (Fig. 1c, Supplementary Table 7).

**The *Red* locus as a candidate regulatory region of *FST***. Gouldian finch head colour is associated with multiple phenotypic traits (see above). As such, the intergenic location of the *Red* locus suggests that this locus acts as a regulatory region that controls a pleiotropic gene. From a small-scale transcriptomic analysis of regenerating feather follicles from the scalp, we identified a list of differentially expressed genes belonging to functionally correlated gene ontology (GO) networks with potential roles in feather development and pigmentation (Supplementary Fig. 3, Supplementary Data 1). Although neither *MOCS2* nor *FST* were differentially expressed between morphs, some genes in this network are known to interact with *FST* (Supplementary Fig. 3, Supplementary Data 1).

The multiple differences in gene expression suggest that the fates of feather precursors are determined by gene(s) that act upstream of the regulatory network at an earlier developmental stage. Previous studies indicate that *FST* is a strong candidate for regulating the colour polymorphism. Follistatin is known to regulate hair and feather development through its antagonistic action on genes of the *TGF-β* superfamily[23,24]. The genes in this family (e.g. *activin*, *TGF-β* and *BMP*) may control melanocyte differentiation and melanogenesis[25,26]. *FST* and its downstream genes might also be responsible for the structural differences in the feathers found in the Gouldian finch polymorphism as *BMP* and its alternative antagonist, *noggin*, are important

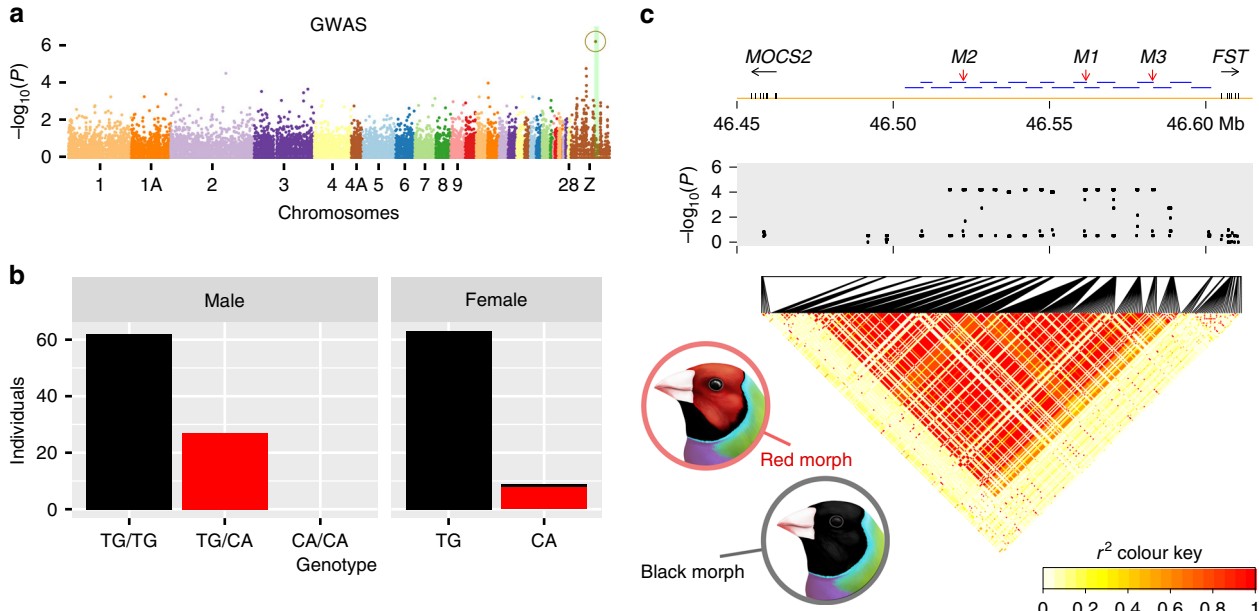

**Fig. 1** Identification of the *Red* locus. **a** Association between RADSeq markers and head colour in Gouldian finches ($n_{black} = 22$, $n_{red} = 10$). A circle around a dot indicates the only significant signal in GWAS ($P < 0.05$, 1000 permutations). The region shaded in green on the Z chromosome represents the candidate region (~7.2 cM) based on a previous linkage mapping study[21], where the *Red* locus co-segregated perfectly with microsatellite markers. **b** Genotypes of 161 birds from a wild population (male: $n_{black} = 62$, $n_{red} = 27$; female: $n_{black} = 64$, $n_{red} = 8$) based on two contiguous SNPs in the *Red* locus, where bar colours represent head colour. These SNPs (TG/CA) were identified by RADSeq (*M1* at 46,561,597–46,561,598 bp in the zebra finch) and were typed using allele-specific PCR (Supplementary Fig. 2, Supplementary Table 2). **c** The extent of linkage disequilibrium (LD) associated with the colour trait. In the top panel, the locations of the protein-coding regions on the zebra finch Z chromosome (Mb) are indicated by vertical bars on the horizontal axis and the horizontal arrows below the gene names indicate the direction of transcription. Horizontal blue bars show the overlapping long-range PCR products (totalling ~100 kbp) that were subsequently sequenced, which includes the 72-kbp region with elevated LD defined as the *Red* locus in this study (see Fig. 2, Supplementary Tables 6, 7). The three vertical red arrows indicate the locations of markers (*M1–M3*) that were used to genotype the wild birds in association tests (Supplementary Fig. 2, Supplementary Table 2). The middle panel shows $-\log_{10}P$ for association tests between phenotype and 282 SNPs from 26 fragments in 16 females ($n_{black} = 8$, $n_{red} = 8$) typed using Sanger sequencing. The lower panel shows pairwise LD ($r^2$) between SNPs. Finch illustrations by Megan Bishop

determinants of the morphogenesis of feather branching[24,27]. The *TGF-β* superfamily and its regulator *FST* are also involved in the development and regulation of the visual[28,29] and reproductive systems[30].

We found no differences in the *FST* protein-coding sequences between black- and red-headed birds, and the amino-acid sequences were conserved between the Gouldian finch and the zebra finch (Supplementary Fig. 4). However, we identified several evolutionarily conserved sites within the *Red* locus located upstream of *FST* (Supplementary Fig. 5), suggesting that this region has functional importance. It is therefore likely that *cis*-regulatory region(s) in the *Red* locus determine spatio-temporal variation in the expression of *FST* that then controls multiple genes in the *TGF-β* superfamily, which in turn may be responsible for the pigmentation difference and other pleiotropic effects. We note that one of the few divergent regions that distinguish the genomes of hybridizing golden-winged (*Vermivora chrysoptera*) and blue-winged (*V. cyanoptera*) warblers has been mapped to the same intergenic region between *MOCS2* and *FST*[31], suggesting convergent evolution of plumage colouration mechanisms and a possibly important role of the *Red* locus as a barrier to gene flow in other taxa. However, the fact that this locus has not led to speciation in the Gouldian finch indicates that selection may instead promote the persistence of divergent morphs in this species.

**The nucleotide diversity and divergence at the *Red* locus.** If selection has played a role in the origin and maintenance of the *Red* locus then it may have left a signature in the pattern of

nucleotide diversity in this region. We calculated point estimates of summary statistics at the *Red* locus, as well as in sliding-windows across the ~100-kbp sequenced candidate region that included the *Red* locus to characterize the pattern of nucleotide diversity and divergence within and between morphs (Fig. 2a–d; Supplementary Table 9). We then compared the *Red* locus with 24 randomly-chosen intronic loci on the Z chromosome, as a putatively neutrally-evolving reference set (Fig. 2e–h; Supplementary Table 9). Notably, diversity for the combined sample of alleles ($Z^R$ and $Z^r$ together) at the *Red* locus is elevated compared with the reference loci, as is the level of differentiation between $Z^R$ and $Z^r$ alleles.

The mean nucleotide diversity at the reference loci ($\pi_{ref}$) was 0.0033 and varied little between the alleles of each morph (Fig. 2e). The point estimate of nucleotide diversity for the combined sample at the *Red* locus ($\pi_{total} = 0.0102$) was above that at all 24 reference loci (Fig. 2a), while diversity within the $Z^R$ ($\pi_{red} = 0.0004$) and the $Z^r$ ($\pi_{black} = 0.0015$) alleles was similar to that at the reference loci.

The estimate of absolute divergence $d_{xy}$ (the mean number of pairwise differences per site[32]) between the $Z^R$ and $Z^r$ alleles at the *Red* locus ($d_{xy\,Red\,locus} = 0.019$) was again greater than between morphs at all 24 reference loci ($d_{xy\,ref} = 0.0033$; Fig. 2b, f) and the estimate of net divergence $d_a$ ($d_{xy}$ minus the mean of the two within-morph $\pi$ values) was not greatly different from $d_{xy}$ due to the low nucleotide diversity within each allelic class (data not shown). In contrast, the divergence between zebra finch and alleles of each morph was similar at both the *Red* locus and the reference loci (Fig. 2b, f).

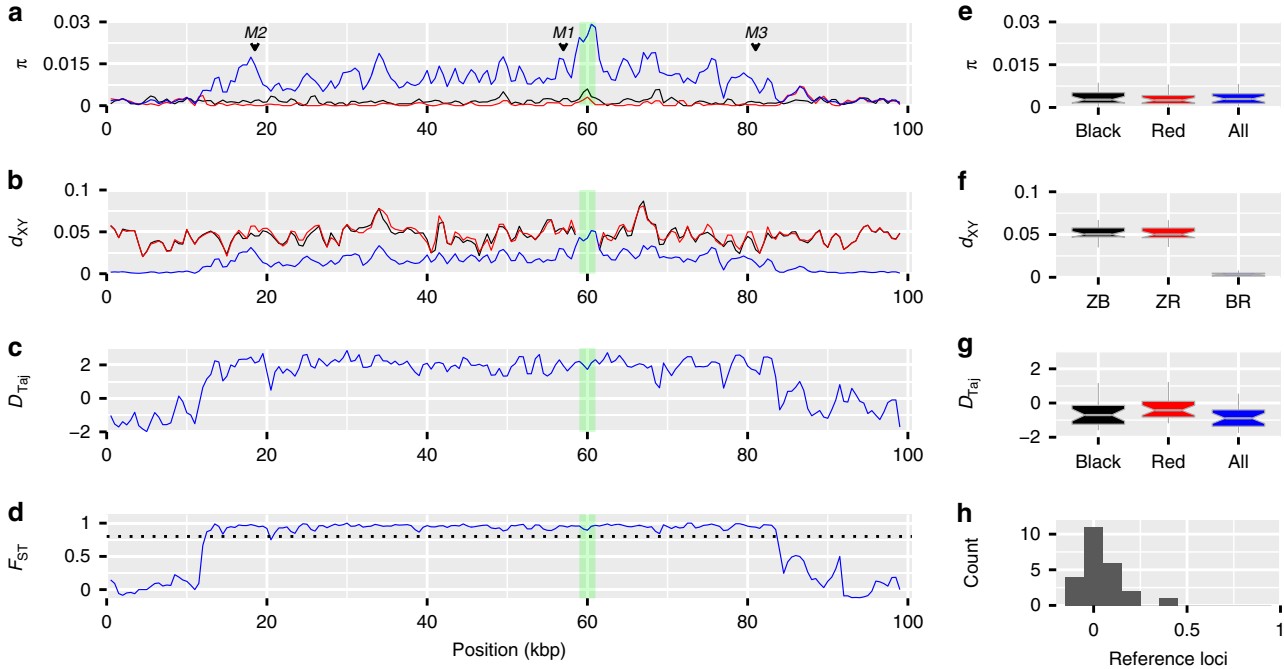

**Fig. 2** Diversity and divergence at the *Red* locus and reference loci. Position in **a–d** represents ~100 kbp alignment of assembled MiSeq reads for 12 females ($n_{black} = 6$, $n_{red} = 6$), which corresponds to the position between 46,503,706–46,601,744 bp in the zebra finch reference genome (the region connected by horizontal blue bars in Fig. 1c). **a–d** Sliding-window analysis (window size of 1000 bp with a step of 500 bp) was used to obtain estimates of summary statistics. Green shading indicates an insertion of a putative LTR retrotransposon in the Gouldian finch that is not found in zebra finch. **a** Nucleotide diversity ($\pi$) of black (black line), red (red line) morphs and all samples combined (blue line). **b** Sequence divergence between zebra finch and each morph (ZB: black line; ZR: red line), and between morphs (BR: blue line). **c** Tajima's D for all samples combined. **d** $F_{ST}$ between morphs. A horizontal dashed line indicates $F_{ST} = 0.8$. The data in **e–h** were based on 24 intronic reference loci on the Z chromosome, with **e–g** showing the same summary statistics as **a–c** and **h** a histogram of $F_{ST}$. The box in **e–g** bounds the interquartile range with a notch showing a 95% confidence interval around the median, and Tukey-style whiskers extend to a maximum of 1.5× interquartile range beyond the box. We define the 72-kbp sequenced region with $F_{ST} > 0.8$ in **d** as the *Red* locus in subsequent analyses

Tajima's $D$[33] was moderately negative at the reference loci, both within the alleles of each morph and with both morphs' alleles combined ($D_{Taj\ black} = -0.599$, $D_{Taj\ red} = -0.304$ and $D_{Taj\ total} = -0.818$), as well as within the alleles of each morph at the *Red* locus ($D_{Taj\ red} = -0.844$ and $D_{Taj\ black} = -0.891$; Fig. 2g, Supplementary Table 9), consistent with population expansion (see Supplementary Methods). However, $D_{Taj}$ was positive in the combined sample at the *Red* locus ($D_{Taj\ total} = 2.193$; Fig. 2c), which is significantly >0 under the standard neutral model ($P_{coalescent} = 0.002$) using simulations in DnaSP v5[34], and again higher than at all 24 reference loci.

In captive populations of the Gouldian finch, pronounced pre- and postzygotic incompatibilities between morphs have been described, and mixed-morph pairings resulted in reduced survival: 40% of male and 80% of female offspring died before 90 days of age[35,36]. If these incompatibilities persisted in the wild, then genome-wide differentiation between the black and red morphs might be expected. To quantify this, we estimated Wright's fixation index $F_{ST}$[37] between each morph's alleles. $F_{ST}$ between alleles of each morph at the *Red* locus reached a near-maximal value ($F_{ST} = 0.951$) and was significantly >0 ($P_{permutation} = 0.003$). This value was greater than $F_{ST}$ between each morph's alleles at all the Z-linked reference loci (composite $F_{ST} = 0.038$, Fig. 2d, h, Supplementary Fig. 1).

In contrast, we found no genomic differentiation with regards to colour morph across the genome outside of the *Red* locus in the wild sample. First, a coancestry matrix[38] constructed for 32 wild birds ($n_{black} = 22$, $n_{red} = 10$) using autosomal RADSeq loci, designed to fully exploit the available haplotypic information, failed to identify any clusters that separated the morphs

(Supplementary Fig. 6). Second, a STRUCTURE[39] analysis between morphs using nine unlinked autosomal microsatellite markers for all the wild birds ($n_{black} = 126$, $n_{red} = 35$) indicated that the most-likely number of subpopulations was one ($K = 1$; Supplementary Fig. 7). These results suggest that there is no or little incompatibility leading to any genome-wide differentiation between morphs in the wild.

**Test for neutrality.** Overall, the patterns of nucleotide diversity and divergence at the *Red* locus (i.e. elevated $\pi$, excess of intermediate frequency alleles and maximal $F_{ST}$) are consistent with balancing selection[40]. However, such patterns can be obtained even under neutral evolution if sampling is conditioned on a site associated with the phenotypes of interest, as in this case (e.g. see the simulated patterns of polymorphism close to the focal site presented in Fig. 3). Therefore, to formally test for neutrality we carried out coalescent simulations under a neutral model that incorporated demography, recombination, and accounted for our sampling scheme and asked if the pattern of genetic variation we observed within the 100-kbp-region is compatible with neutral expectations (see Methods and Supplementary Methods, Supplementary Figs. 8–17, Supplementary Tables 11–14 for detailed procedures of the neutrality test).

Our simulations show that the observed patterns of genetic variation at the *Red* locus cannot be explained by either neutral processes or biases caused by the sampling scheme (Fig. 3, Supplementary Methods). Assuming the $Z^R$ allele is derived (i.e., the derived allele frequency, DAF = 0.144), the observed nucleotide diversity ($\pi_{total}$) and Tajima's D for the combined sample of

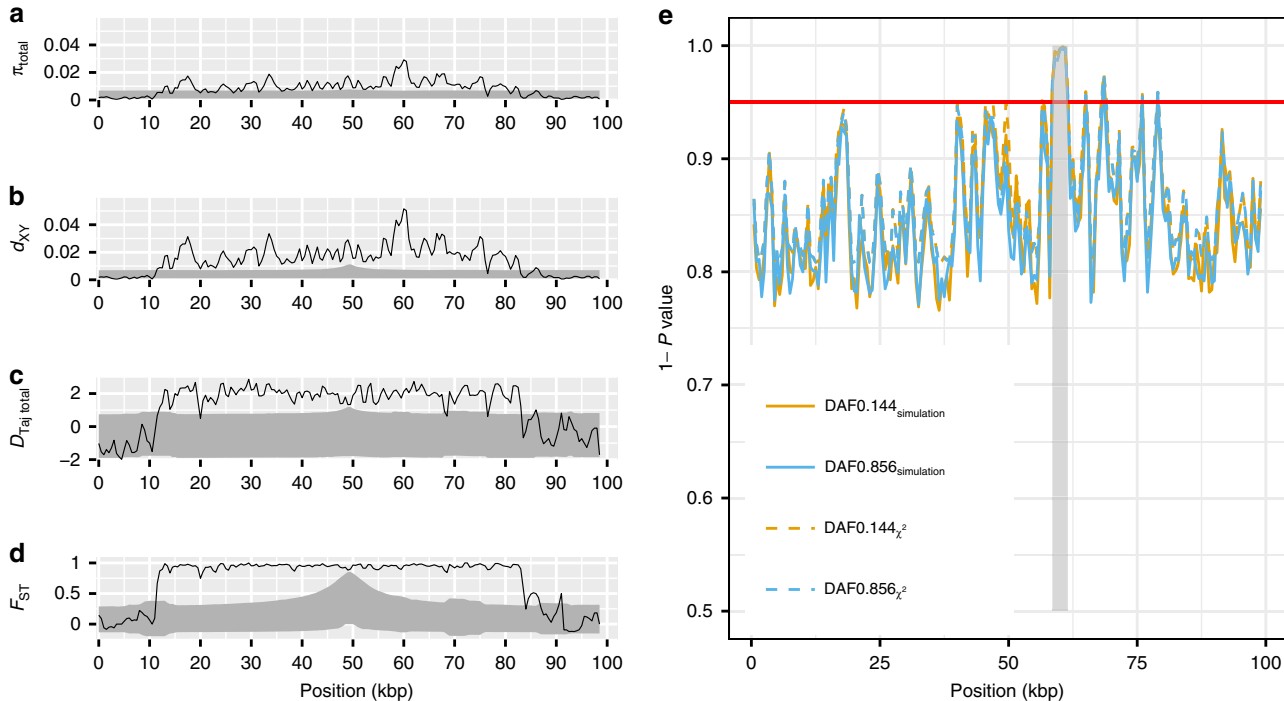

**Fig. 3** Test for neutrality using coalescent simulations. Position in **a**–**e** represents ~100 kbp alignment of assembled MiSeq reads for 12 females ($n_{black}$ = 6, $n_{red}$ = 6) as in Fig. 2. **a**–**d** Shown are sliding-window comparisons of the observed polymorphism at the *Red* locus with the polymorphism simulated under a neutral model that incorporated the sampling scheme (six alleles for each of the red and black morphs), demography, recombination, and a derived allele frequency (DAF) of 0.144 at the focal site (see Supplementary Methods). Solid lines represent the observed data (cf. Fig. 2) and the grey-shaded regions represent the space encompassed by the 95% confidence intervals derived from the simulations. **a** Nucleotide diversity for both morphs' alleles combined. **b** $d_{XY}$ between the two morphs' alleles. **c** Tajima's *D* for both morphs' alleles combined. **d** $F_{ST}$ between the two morphs' alleles across the *Red* locus. **e** A sliding-window implementation of the modified HKA test, taking into account the observed sampling scheme, allele frequencies, mutation rate and recombination rate. One minus the *P*-value is plotted for each window for DAF of 0.144 (yellow lines) and 0.856 (blue lines), using the chi-squared distribution (dashed lines) and simulations (solid lines). The red horizontal line indicates a *P*-value threshold of 0.05. The shaded grey bar indicates an insertion of a putative LTR retrotransposon in the Gouldian finch that is not found in zebra finch

$Z^R$ and $Z^r$ alleles at the *Red* locus ($D_{Taj\ total}$) both lie above the 95% confidence intervals (CIs) of the simulations for most windows, as do $F_{ST}$ and $d_{xy}$ between the $Z^R$ and $Z^r$ alleles (Fig. 3a–d). In contrast, statistics calculated within the $Z^R$ or $Z^r$ alleles fell within the 95% CIs of the simulated distributions, with the exception of nucleotide diversity within the $Z^R$ allele ($\pi_{red}$, Supplementary Fig. 9). The same pattern was observed when we assumed that the $Z^r$ allele was derived (DAF = 0.856, Supplementary Fig. 10). In particular, the observed region of greater than expected differentiation extended over an ~72-kbp interval, with a rapid drop on either side—contrasting with neutral expectations for a single site on which sampling is conditioned, which results in a gradual decline towards background levels over a 20-kbp interval (Fig. 3d). As we have demonstrated here, there is no evidence of an inversion across this region (Supplementary Table 7). Moreover, we found evidence for the presence of recombination within the *Red* locus (Supplementary Table 13, Supplementary Methods). Therefore, the highly diverged haplotype blocks are not likely to be maintained by reduced recombination, but rather by selection against recombinants. A sliding-window implementation of the HKA test[41], taking into account nonrandom sampling, demography and recombination at the *Red* locus, also revealed significant departures from neutrality for multiple windows across the locus (Fig. 3e).

**The evolutionary history of the black and red haplotypes.** Balancing selection can maintain polymorphism over extended periods of time, resulting in deeply diverged haplotypes that may

be even older than the species in which they reside[40]. In a gene genealogy of the M1 region (see Fig. 1c) for 14 species in the family *Estrildidae*, sequence divergence between black and red haplotypes was deeper than that between other species or even between genera (Fig. 4, Supplementary Fig. 18). Four other *Erythrura* species formed a monophyletic group (bootstrap support 100%), separate from the Gouldian finch, whereas none of the Gouldian finch haplotypes clustered with the sequences of red-faced finches (*E. pealii* and *E. psittacea*) or blue-faced finches (*E. tricolor* and *E. trichroa*). Despite the high divergence, the two Gouldian finch haplotypes formed a clear monophyletic group (bootstrap support 100%), distinct from other *Erythrura* species, suggesting that these haplotypes evolved within the Gouldian finch lineage (Fig. 4). These patterns suggest that the divergent haplotypes at the *Red* locus are old (nearly half of the divergence between zebra finch and the Gouldian finch), although we did not attempt to estimate the age since balancing selection is likely to affect this estimation. Near the highly divergent M1 region, we identified an insertion with the characteristic feature of a long terminal repeat (LTR) retrotransposon, which is absent in the zebra finch (Fig. 2). Transposable elements are often associated with adaptive polymorphism[42]. Indeed, signals of selection, including the departure of the observed data from expectations under neutrality derived from simulations as well as from the HKA test, were strongest in this region (Fig. 3e, Supplementary Methods). However, comparison of the 3′ and 5′ LTRs in each haplotype suggests that the divergence between the colour-associated haplotypes did not occur until after the

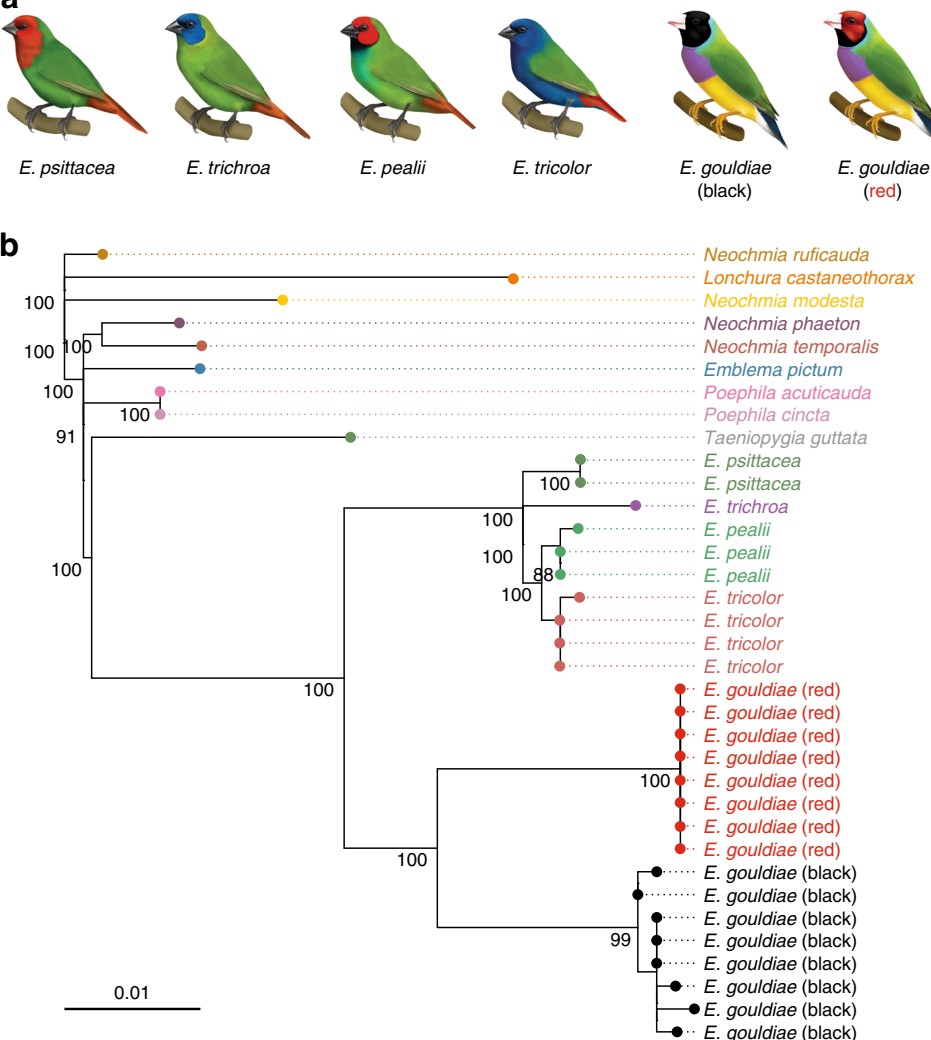

**Fig. 4** Gene genealogy at the *Red* locus. **a** Illustrations of species in the genus *Erythrura*. From left, *E. psittacea*, *E. trichroa*, *E. pealii*, *E. tricolor*, and black and red morph of *E. gouldiae* (illustrations by Megan Bishop). **b** Maximum-likelihood tree (GTR+Γ+I) for the species in the family *Estrildidae* using 774-bp alignment of *M1* (Fig. 1). Node support values (%) were generated from 1000 bootstrap replicates and values >80% are shown next to the branches

insertion of the retrotransposon, which occurred some time after the divergence of Gouldian finch and zebra finch (Supplementary Fig. 19).

In a relative rate test, the red haplotype showed disproportionately longer branch lengths than the black haplotype across the *Red* locus (Supplementary Figs. 18 and 20), suggesting that the two haplotypes have evolved with unequal evolutionary rates. Although the driver of this asymmetry remains unclear, one possible explanation is that the red haplotype may have undergone more rounds of adaptive substitution, which would also explain the lower diversity of the red haplotype over a long stretch (Supplementary Figs. 9b and 10f).

**The origin of divergence and the mode of balancing selection.** There are several hypotheses for the origin of haplotypic divergence at the *Red* locus, including former allopatry followed by secondary contact[43] or the lateral transfer of ancestral variation[44]. For example, in northern Australia, the historical Ord Arid Intrusion event resulted in the splitting of many co-distributed taxa into three distinct biogeographical populations, and some became (and remain) distinct subspecies[45].

Regardless of the initial cause of divergence, our analyses suggest that balancing selection may be responsible for maintaining the divergent haplotypes at the *Red* locus. There are a few possible mechanisms of balancing selection in this species. First, red males may have an advantage under sexual selection, as females of all morphs of the Gouldian finch show a preference for red males[35]. Moreover, the dominant status of red males in the social structure[46] may be beneficial when competing for limited resources, such as nest sites, albeit at the expense of reduced paternal care[47]. Since the red allele is genetically dominant, its allele frequency would be expected to quickly rise to a high level under such a selective advantage. However, the red phenotype is present at low frequency in natural populations[12]. Therefore, opposing selection has probably prevented the spread of the red allele in the population. Red males suffer from high stress levels and have significantly poorer reproductive outcomes than black birds when aggressive red males are in high frequency, and are consequently in a more competitive and stressful environment[48–50]. Thus, frequency-dependent selection may have acted to maintain a low equilibrium frequency of the red individuals. Alternatively, this may be an example of sexual antagonism[51], in which the fitness effects of different colouration

are not the same in both sexes, and selection against red females prevents the red haplotype from increasing to a high frequency, despite being favoured in males. Red females fully express red head colour in captive populations (as a result of strong artificial selection by aviculturists). However, those in the wild population are less brightly coloured generally; the proportion of red feathers on their heads is typically lower, and they are often indistinguishable from black females[9], suggesting selection against redness in females in the wild.

In this study, we have shown that a single locus upstream of *FST* is associated with colouration. This locus is intergenic, which does not support a role for a supergene in the classical sense in differentiating the red and black colour morphs of Gouldian finches. But it provides an example of a putative regulatory element with downstream effects on multiple phenotypes being maintained in the long-term by balancing selection. This has resulted in the maintenance of markedly divergent morphs, despite gene flow within the species, and in the absence of genomic rearrangements.

During the revision of the current manuscript we were alerted to a study on face colour polymorphism in domesticated Gouldian finches by Toomey et al.[52]. The research presented here confirms, complements, and extends that detailed study in several ways. First, both independent approaches highlight the same genomic region upstream of the *follistatin* gene, providing strong evidence of its functional role in generating the colour polymorphism. Second, unlike Toomey et al.[52], our analysis incorporates data from wild birds, making clear the relevance of this region to non-domesticated populations. Finally, our simulations and evolutionary reconstructions add key historical context to how and why this polymorphism has evolved and is maintained in the wild.

The lack of wider genomic divergence between morphs suggests that the *Red* locus may not, alone, be sufficient to induce the incompatibilities observed in captive populations. This, in turn, suggests that the extended LD in captive populations (e.g. an ~8 Mb region showing zero recombination around the *Red* locus[21]) might encompass other linked loci underlying these incompatibilities. Therefore, further studies of the region around the *Red* locus will provide an important opportunity to address novel questions about the evolution of colour polymorphism. In particular, understanding the functional role of *Red*, a pleiotropic switch gene associated with both melanin and carotenoid colouration, may shed light on the mechanistic basis of sexual selection and genetic incompatibilities between colour morphs, which are both key evolutionary factors in speciation[53].

## Methods

**DNA samples**. Blood samples were collected from 161 Gouldian finches (male: $n_{black} = 62$, $n_{red} = 27$; female: $n_{black} = 64$, $n_{red} = 8$) in 2008 from a single wild population in Western Australia (within a 50-km radius of Wyndham in East Kimberley). Birds were either caught in mist nets at waterholes or at their nests. The population size was estimated to be around 300 adults. Genomic DNA was extracted using the ammonium acetate technique. The sex of each sample was determined using two sex-linked markers, *Z-002D*[54] and *Z37B*[55] that amplify alleles on both the Z and W chromosomes. No power analysis was performed prior to the study. All 161 samples were used for the association test and a population structure analysis. A subset of samples was randomly selected and used for RADSeq and subsequent Sanger and MiSeq sequencing and genotyping (see below). Throughout the study, traits were measured blind to treatment and to values of other traits.

For whole-genome sequencing (WGS), blood samples were collected from 24 Gouldian finches (males: $n_{black} = 4$, $n_{red} = 4$, $n_{yellow} = 3$; females: $n_{black} = 3$, $n_{red} = 5$, $n_{yellow} = 5$) that were sampled from a single captive population of domesticated birds in Ithaca, NY, in 2016. Yellow birds have the same allele at the *Red* locus as the red morph but develop as yellow under the control of an unknown epistatic autosomal locus[11]. Genomic DNA was extracted from these blood samples using DNeasy kits (tissue protocol; Qiagen, Hilden, Germany).

**RADSeq**. We genotyped 32 birds from the wild population described above using RADSeq[20] in two separate batches. In the first batch, DNA fragment libraries for 20 individuals ($n_{black\_male} = 10$, $n_{red\_female} = 8$ and $n_{red\_male} = 2$) were constructed following restriction digestion with *Sbf*I, and the pooled libraries were sequenced using two lanes of Illumina Genome Analyser IIx. This sampling scheme was to identify the *Red* locus efficiently with limited sample size in GWAS (see below). In the second batch, a pooled library, including 12 additional black males from the same population, was constructed using the same protocol but sequenced on an Illumina HiSeq2000 instrument.

The quality of the short sequence reads ($2 \times 100$ bp) was checked using FASTQC (http://www.bioinformatics.babraham.ac.uk/projects/fastqc/). We cleaned and deduplicated the sequence reads using the *process_radtags* and *clone_filter* utility programmes from STACKS[56]. We then mapped the sequence reads against the zebra finch (*Taeniopygia guttata*) reference genome assembly (taeGut3.2.4) using SeqMan NGen 4.0 (DNASTAR, Madison, Wisconsin, USA), allowing up to 15% divergence between sequence reads and the reference sequence. Single-nucleotide variants were called using SAMTOOLS v1.2[57] and filtered based on the minimum depth for each site per individual (>2), minimum SNP quality (>250), maximum of missing genotypes across all individuals (<30%) and minor allele frequency filter (>0.05) using custom R scripts and VCFtools v0.1.12[58]. Although the HiSeq2000 sequencing runs had more coverage than the GAIIx runs, by applying filtering based on missing genotypes we could effectively remove any sites that were found only in the HiSeq2000 run. Mean depth per locus in an individual was different among libraries (GAIIx: L1 = 3.2, L2 = 7.8; HiSeq2000: L3 = 13.2). However, because we mixed equal numbers of black and red birds in L1 and L2, any bias due to the coverage in the inference of structure between morphs should be limited. Females were encoded to have only one allele for the genotypes of Z-linked loci using the *vcf-fix-ploidy* function of VCFtools. In an $F_{ST}$ outlier analysis (Supplementary Fig. 1), we included 32 individuals ($n_{black\_male} = 22$, $n_{red\_female} = 8$ and $n_{red\_male} = 2$) whose genotypes at the *Red* locus were either homozygous or hemizygous, except for the two heterozygous red males.

**GWAS**. SNPs linked to the causative site at the *Red* locus are expected to show a recessive homozygous genotype in black males ($a/a$), to be hemizygous ($A$) in red females and to be dominant homozygotes ($A/A$) or heterozygotes ($A/a$) in red males (see Introduction). In a genome-wide association study (GWAS) in 32 RAD-sequenced samples, we used Fisher's exact tests with one degree of freedom for a dominant model of penetrance in a $2 \times 2$ contingency table of phenotype (black or red) by genotype ($a/a$ or not $a/a$) using PLINK v1.9[59] (Fig. 1a). We treated the hemizygous Z-chromosome genotypes of females as homozygotes. We corrected for genomic structure among samples using the *--within* option and the family-wise estimate of the SNP's significance was determined and corrected for multiple testing using the *--mperm* option (1000 permutations).

**De novo low-coverage whole-genome sequencing**. Sequencing libraries for the 24 captive Gouldian finches were generated using the Illumina TruSEQ PCR-free Library Preparation Kit following the protocol for generating 350-bp libraries. These 24 individuals were sequenced on a single lane of an Illumina NextSeq500 instrument to produce 150-bp paired-end reads. We used AdapterRemoval[60] to collapse overlapping paired reads, trim sequences of "Ns" along the 5′ and 3′ ends of reads, and remove sequences shorter than 20 bp. This resulted in a mean of 33.7 million total reads per individual (range 20.8–49.3 million reads). We used Bowtie 2[61] to map each of the individual reads to the zebra finch genome to a mean coverage of ~3.6×, using the "very sensitive local" set of alignment presets. For SNP discovery and variant calling, we used the UnifiedGenotyper in GATK[62]. We removed possible variants that had "quality by depth" (QD) of <2 and "mapping quality" (MQ) of <30 (the filtering conditions were QD < 2.0, FisherStrand >40.0, MQ < 30.0, HaplotypeScore >12.0, MappingQualityRankSum < −12.5, Read-PosRankSum < −8.0). Variant confidence is a measure of sequencing depth at a given variant site; mapping quality refers to the root-mean-square of the mapping confidence (from Bowtie 2) of reads across all samples. We applied additional filters using VCFtools. First, we coded genotypes with a Phred-scaled quality lower than 8 as missing data. This is a permissive filter, necessary given our low-coverage data, which allows for genotypic calls with as few as three reads at a given site for an individual. We excluded loci with more than 30% missing data and used a minor allele frequency filter of 5%. This resulted in 4,818,133 SNPs distributed across the genome.

To properly correct for the ploidy of genotypes identified on the Z chromosome, we used the *vcf-fix-ploidy* function of VCFtools to assign one allele at each site for Z-linked loci in females. To increase the chance of detecting the location of the *Red* locus in an $F_{ST}$ outlier analysis (Supplementary Fig. 1), we only included data from 17 individuals whose genotypes were known to be homozygous or hemizygous for alternative alleles at the *Red* locus based on the pedigree (black: $n_{black\_male} = 4$, $n_{black\_female} = 3$; red: $n_{red\_female} = 5$ and $n_{yellow\_female} = 5$).

**Test for the association with colour phenotype**. Several segregating sites associated with head colour were identified in the sequences around a RADSeq locus we found to be associated with head colour (locus *Ego165*, Supplementary Table 4). To confirm the association with plumage colour, 161 birds were genotyped for

alternative two-base pair variants (TG/CA) using an allele-specific PCR with the Multiplex PCR Kit (QIAGEN) and a set of primers (locus *M1*, see Supplementary Table 2). The conditions for the PCR were 95 °C for 15 min, 30 cycles of 94 °C for 30 s, 64 °C for 90 s and 72 °C for 1 min, followed by 72 °C for 5 min. The PCR products were separated on an ABI 3730 DNA Analyser (Applied Biosystems, Foster City, California, USA) and scored using GENEMAPPER v 3.7 (Applied Biosystems). The association of the genotypes with face colour was tested for each sex using Fisher's exact tests in R v3.2.5 (R Core Team, 2015, Fig. 1b). In addition to the locus *M1*, a multiplex marker set was developed including two allele-specific markers to genotype loci at the edges of the region of elevated LD at the *Red* locus (*M2* and *M3*, Supplementary Table 2, Supplementary Fig. 2). The same condition as above was used in the PCR, except for an annealing temperature of 60 °C. We examined the deviation of the *M1* genotypes from Hardy–Weinberg equilibrium (HWE) using Graffelman–Weir's exact test for sex-linked markers implemented in the HardyWeinberg package[63] and obtained the *P*-value by permuting individual genotypes 10,000 times (Supplementary Table 3).

**The extent of LD and sequencing of a candidate gene**. To determine the extent of LD across the *Red* locus and the association of marker loci with the colour trait, the DNA sequences of 26 short fragments in the upstream region and the whole exon of an adjacent candidate gene (*follistatin*) were obtained (Supplementary Table 4). We sequenced an additional 15 loci on the Z chromosome that are not close to the *Red* locus as controls (Supplementary Table 4). Genomic DNA from 16 females ($n_{black} = 8$, $n_{red} = 8$) was amplified and cleaned with ExoSAP-IT (USB, Cleveland, Ohio, USA). After Sanger sequencing with the BigDye Terminator v 3.1 cycle sequencing kit (Applied Biosystems), and precipitation using the ethanol/EDTA method, the products were separated on an ABI 3730 DNA Analyser and the bases were called using CodonCode Aligner v2.0.6 (CodonCode Corporation). Association tests between phenotype and 282 SNPs from 26 fragments, and the estimation of pairwise LD between SNPs, were performed using Haploview v4.2[64] and visualized using LDheatmap[65] (Fig. 1c). The coding sequences of *follistatin* in 16 females were aligned, along with the homologous assembled zebra finch sequence, to examine the differences between species and morphs (Supplementary Fig. 4).

**Test for the presence of an inversion**. If there is an inversion breakpoint between two neighbouring loci, then the inverted chromosome will not produce amplicons across the breakpoint in a PCR that produces an amplicon on the non-inverted chromosome. To test for an inversion across the *Red* locus, 19 contiguous, over-lapping fragments were amplified in 12 females ($n_{black} = 6$, $n_{red} = 6$) from the wild population by combining each forward primer with a reverse primer designed within the adjacent amplicon (Supplementary Table 6) and using the LongRange PCR kit (QIAGEN). When the primer sets originally designed from zebra finch sequences failed to produce long amplicons, we designed new primers using the Gouldian finch sequences obtained from previous sequencing efforts to ensure the failure was not due to mismatching sequences in the primer-binding sites (Supplementary Table 6). We confirmed that the long fragments amplified the correct sites by sequencing the boundaries of each long fragment using PCR primers and comparing with the zebra finch reference genome.

Long amplicons across the *Red* locus from the above PCR experiments and short amplicons for the reference loci (see below) were labelled by individual, pooled and sequenced using the Illumina MiSeq platform. All traces of adapter sequences added to amplicons during paired-end short-read (250-bp) sequencing were removed using a custom Perl script (TagCle v0.80, available upon request from K.-W.K.). De novo assemblies were performed for each individual using NGen. Supercontigs were produced by BLAST searching the contig sequences against the zebra finch reference sequence and comparing the positions of the contigs and the overlapping sequences. The assembled sequences were aligned using UGENE v1.12.3[66] and manually edited using BioEdit v7.1.11[67]. These data were used to measure DNA polymorphism across the *Red* locus and in coalescent simulations to test for selection (see below).

**Transcriptome analysis**. We tested for differential gene expression in the feather follicles between the Gouldian finch morphs. First, we sampled regenerating head feathers from birds held by breeders in Australia during the annual moulting period and stored them directly in RNAlater (Qiagen). The feathers were subsequently frozen solid with liquid nitrogen, homogenized using a pestle and mortar, and RNA was extracted using TRIzol (Thermo Fisher Scientific, Waltham, Massachusetts, USA) and RNeasy Mini kits (Qiagen). The quality of the RNA was measured on a Bioanalyzer (Agilent, Santa Clara, California, USA) and high-quality samples with an RNA integrity value > 7 were used in microarray analysis. We obtained the gene expression profiles of six birds ($n_{black\_male} = 3$, $n_{red\_male} = 2$, $n_{red\_female} = 1$) using the Affymetrix zebra finch gene array ST1.0 that contains probe sets from 18,595 genes. Gene expression was normalized using a robust multi-array average algorithm implemented in the Affymetrix Expression Console v1.3. The expression differences were tested using one-way ANOVA implemented in Transcriptome Analysis Console v2.0 (Affymetrix). Because of the small sample size, no gene passed the threshold required to correct for the false discovery rate. Therefore, we used uncorrected *P*-values <0.05 and log$_2$-fold changes ≥1.3 to select

304 potentially differentially expressed (DE) genes. Human orthologues of DE genes were obtained from UniProt (www.uniprot.org). A functionally grouped annotation network for the DE genes was constructed using the Cytoscape plugin ClueGO v2.3.5[68]. Functionally related Gene Ontology (GO) terms for biological processes in human (23 February 2017) were grouped based on a kappa score >0.4 with network specificity of 3–5. The significance was examined using a two-sided hypergeometric test and the false discovery rate was corrected using the Benjamini–Hochberg method[69].

**Evolutionarily conserved elements within the *Red* locus**. We used PhastCons values[70] to examine the presence of the evolutionarily conserved elements within the *Red* locus that might represent regulatory regions potentially responsible for the expression of a candidate gene underlying colour polymorphism. The position of human and mouse transcripts for *MOCS2* and *FST*, and the pre-calculated PhastCons values across the *Red* locus, were obtained from the UCSC Genome Browser Gateway (https://genome.ucsc.edu/cgi-bin/hgGateway). PhastCons values were based on multi-genome alignments among three vertebrates; medium ground finch (*Geospiza fortis*), human (*Homo sapiens*) and mouse (*Mus musculus*), and among five birds: medium ground finch, zebra finch, budgerigar (*Melopsittacus undulatus*), chicken (*Gallus gallus*) and turkey (*Meleagris gallopavo*). The records were visualized using *Gviz*[71]. The genomic position of the *Red* locus in the genome of the medium ground finch was determined using a BLAT search in the UCSC.

**Genomic regions under consideration in the neutrality test**. We obtained a 99,669-bp stretch of sequence using the Illumina MiSeq (see above) encompassing the *Red* locus (46,503,706–46,601,744 bp in zebra finch, Supplementary Table 6) in 12 female birds ($n_{black} = 6$, $n_{red} = 6$) from the wild population. Because females are hemizygous at the *Red* locus, these sequences represent six each of the black ($Z^r$) and red ($Z^R$) alleles.

In an additional MiSeq run, we also randomly chose 24 loci (henceforth the 'reference' loci, Supplementary Table 8) across the Z chromosome, outside of the LD block of the *Red* locus, to measure the background level of diversity for comparison with that seen at the *Red* locus. These loci consisted of single introns, sampled in 11–12 female individuals, 10 of which (5 each of black and red) were also represented in our sample of 12 birds sequenced at the *Red* locus.

**Estimates of DNA polymorphism and divergence**. In order to compare diversity at the *Red* locus (combining the $Z^r$ and $Z^R$ alleles) with that at the reference loci using the MiSeq data, we calculated $\pi$ and Tajima's $D$ for the whole sample at each locus using a custom R script. We then calculated the same statistics within each allelic class (we define two allelic classes: one each for birds having the $Z^r$ or the $Z^R$ allele at the *Red* locus, respectively) at the reference loci and the *Red* locus, in order to compare polymorphism for the combined sample with that within each allelic class. In addition, we calculated $F_{ST}$ and $D_{XY}$ between the two allelic classes at the reference and the *Red* loci using custom R and Python scripts. We used Weir and Cockerham's definition of $F_{ST}$[37], and the estimator provided by Hudson et al.[72] to calculate site-by-site $F_{ST}$ values (Supplementary Fig. 1). To obtain a composite estimate of $F_{ST}$ from multiple SNPs for each window, we used equation (6) of Jackson et al.[73]. In the estimation of $F_{ST}$, $D_{XY}$ and $\pi$ across the candidate region using sliding-window analyses (window size of 1000 bp with a step of 500 bp; Figs. 2 and 3), we retained sites with missing data to maximise the number of SNPs in each window. However, because Tajima's $D$ cannot be calculated when data are missing, we removed individuals with missing data on a *per window* basis for this statistic.

We define statistics that refer to subsets of the data using subscripts: for example, $\pi_{red}$ refers to the nucleotide diversity among the alleles of red birds, and $\pi_{total}$ refers to nucleotide diversity among the alleles of both black and red birds combined. When conducting point estimates of summary statistics *per locus* we removed sites with missing data in any individual. To test for the statistical significance of the point estimate of Tajima's $D$, we used a coalescent simulation under the standard neutral model implemented in DnaSP v5[34].

To examine the genetic structure with regards to colour polymorphism, we constructed a coancestry matrix for samples for which we had RADSeq data that summarises the nearest-neighbour haplotype relationship and provides a more sensitive inference on population structure than the STRUCTURE and PCA approaches[38,44] (Supplementary Fig. 6). We selected autosomal RADSeq loci with ≤5 SNPs (total 18,719 RADSeq loci) and individuals with ≤20% missing data. We used fineRADstructure[38] to obtain coancestry coefficients between individuals and data missingness for each individual. Two individuals had significantly high levels of missing data and were excluded from calculation of the coancestry matrix (Supplementary Fig. 6).

To examine the differentiation between morphs in all wild-population individuals, we used nine autosomal microsatellite markers from a set of previously-tested cross-species markers[55] (Supplementary Table 10) and genotyped 161 birds ($n_{black} = 126$ and $n_{red} = 35$) using the ABI3730 DNA Analyser. Deviation of microsatellite genotypes from HWE was examined using CERVUS v3.0.3[74]. A model-based clustering method implemented in STRUCTURE v2.3.4[39] was used to infer the number of subpopulations ($K$) and to assign individuals to populations

(length of burnin period = 500,000; number of MCMC repeats after burnin = 750,000; 20 iterations for each run, Supplementary Fig. 7).

**Considering nonrandom sampling in the test for selection.** It is important to take into account nonrandom sampling with respect to the *Red* locus in simulations to generate polymorphism data under the null hypothesis of neutrality because, even under neutrality, it is possible to obtain high $F_{ST}$ values, as well as elevated $\pi$ at a focal site, if the sampling scheme is conditioned on that site, as was the case here. In order to replicate our observed sampling scheme for the purpose of statistical testing, we used the allele frequencies from the genotypes of 161 wild birds at the *Red* locus for this purpose (Supplementary Table 3). As we do not know which allele ($Z^r$ or $Z^R$) is ancestral, it was necessary to consider both scenarios in turn; i.e. $Z^r$ is derived (derived allele frequency, DAF = 0.856); or $Z^R$ is derived (DAF = 0.144) (see Supplementary Methods for details about allele frequency estimation). This is, in part, because we cannot be sure which site(s) within the *Red* locus is causal, and so cannot polarise that site using an outgroup sequence. In addition, we lack outgroup sequence data for a portion of the *Red* locus that coincides with a putative TE insertion in the Gouldian finch lineage, but which does not exist at the same location in the zebra finch genome. As such, for the majority of the simulations and analyses below, we arbitrarily set the focal site at the centre of the *Red* locus. However, our results are robust to different choices of this parameter (see Supplementary Methods).

**Testing for selection.** We asked if the patterns of genetic variation we observed within the 100-kbp region are compatible with neutral evolution, given the fact that variants underlying the trait are in this region, and that our sampling is nonrandom. We employed two widely-used methods to test for evidence of natural selection. First, we used coalescent simulations to generate polymorphism data under neutrality at a locus that is equivalent to the ~100-kbp sequenced region in Gouldian finches in terms of its allele frequencies in nature (see above), the sampling scheme (6 each of red and black birds) and relevant population genetic parameters (the population mutation rate $\theta = 0.0144$ estimated from 24 reference loci, and the population recombination rate $\rho = 0.004$ estimated from the ~100-kbp sequenced region). Note that the simulations based on the 100-kbp region are robust to the estimate of the recombination rate because significant differentiation was also apparent between the red and black alleles in a simulation using $\rho = 0.0002$, which is 20× lower than the estimate from the 100-kbp sequenced region ($\rho = 0.004$) (Supplementary Figs. 11 and 12). The lower value of $\rho$ approximates that inferred by LDhat[75], both within the red and the black allelic classes in the 100-kbp region. It should be noted that the strong LD between the alleles prevented us from using LDhat to assay recombination in the combined sample. The conclusions from the simulations were also robust to limiting the range of sequence simulated to the 72-kbp *Red* locus (Supplementary Figs. 13 and 14). The estimate of $\rho$ from this region was unchanged from the value for the entire sequenced region. By assessing whether the observed polymorphism in the sequenced region is compatible with these simulations, we can accept or reject the null hypothesis of neutrality. We used the programmes SelSim[76] and mbs[77] to simulate polymorphism linked to a focal site at which a mutation has arisen and spread by genetic drift to the frequencies we observed at the *Red* locus (0.144 or 0.856). These programmes allowed us to replicate the allele frequencies, sampling scheme and population genetic parameters (see above) observed at the *Red* locus and obtain expectations under neutrality. In addition, mbs can model demographic changes. We also used a modified version of the HKA test[41] to test for selection. See Supplementary Methods for more detailed methods.

**Gene genealogy and relative rate test.** To infer the evolutionary history of the haplotypes at the *Red* locus we constructed gene genealogies and compared the level of evolutionary divergence between the black and red haplotypes with those of orthologous sequences in other species. First, blood samples of 14 species in the family Estrildidae including red-faced (*E. pealii* or *E. psittacae*) or blue-faced (*E. tricolor* and *E. trichroa*) *Erythrura* species were obtained from an aviculturist. Blood samples were collected and stored using Whatman FTA cards (GE Healthcare, Little Chalfont, UK) and DNA was extracted following the manufacturer's instructions. Sequences of the fragments amplifiable using the *M1–M3* primers (Supplementary Table 2) were obtained using Sanger sequencing. For zebra finch, the sequences from the reference genome assembly were used.

We tested alternative nucleotide substitution models using the R package, *phangorn* v2.3.1[78], which suggested GTR+Γ+I to be the best model for the sequence of the *Red* locus. Maximum-likelihood trees were then constructed using this model and tested with 1000 bootstraps using *phangorn* and visualised using *ggtree* v1.4.20[79]. To examine if the molecular clock hypothesis applied to the *Red* locus, we used Tajima's nonparametric relative rate test implemented in pegas v0.10[80].

**Compliance with ethical standards.** Blood sampling of birds was conducted under the approval of the Animal Ethics Committees at the University of New South Wales and Macquarie University (ARAs 2007/037 & 2007/038).

**Reporting summary.** Further information on research design is available in the Nature Research Reporting Summary linked to this article.

## Data availability
The data that support the findings of this study are available from the corresponding authors on request. PLINK files for RADSeq analysis and sequence alignments of de novo sequencing for the *Red* locus underlying Figs. 1a and 4b, Supplementary Fig. 1, 2, 6 and 8–19 and Supplementary Tables 9 and 11–13 are available from the Dryad Digital Repository [https://doi.org/10.5061/dryad.9nk3757]. Raw Illumina reads are available at NCBI SRA (SAMN10751906-SAMN10751929) under BioProject PRJNA515277. Microarray data are available on the Gene Expression Omnibus (GEO) under accession number GSE125295. A reporting summary for this article is available as a Supplementary Information file.

## Code availability
Custom scripts are available from the corresponding authors on request.

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

## Acknowledgements

We thank Brian and Deborah Charlesworth for comments on an earlier version of the manuscript. Sarah Pryke and David Harris provided blood samples of the Gouldian finches and other species of *Estrildidae*. Peri Bolton assisted with the collection of cheek feathers. We thank the Natural Environment Research Council Biomolecular Analysis Facility (NBAF) at the University of Edinburgh for sequencing RAD libraries (NBAF674) and NBAF at the University of Liverpool for gene expression microarray scanning. This study was supported by a University of Sheffield scholarship to K.-W.K., a joint University of Sheffield/Natural Environmental Research Council (NE/H524881/1, NE/K500914/1) Ph.D. studentship to B.C.J., a UK Department for Business, Innovation and Skills/Ministry of Education of China UK–China Scholarships for Excellence studentship to H.Z., an NSERC Banting postdoctoral fellowship to D.P.L.T., an Australian Research Council (DP130100418) grant to S.C.G. and T.B., and a Leverhulme Fellowship to T.B.

## Author contributions

K.-W.K., B.C.J., S.C.G., K.Z. and T.B. conceived and designed the study. S.C.G. managed the work on Gouldian finches in the wild in Australia. K.-W.K., M.M.L. and A.D. constructed the RAD library. D.P.L.T., S.A.T., E.I.G and I.J.L. performed whole-genome sequencing. K.-W.K. and H.Z. performed molecular work. K.-W.K. and B.C.J. analysed the data. K.-W.K. and B.C.J. wrote the paper with contributions from all other authors.

## Additional information

**Competing interests:** The authors declare no competing interests.

