## [Peer Review File · Nature Communications]

Reviewers' Comments:

Reviewer #1:

Remarks to the Author:

This is an interesting study on a fascinating bird species with a color polymorphism that is also associated with a suite of behavioral and physiological traits. The authors present a somewhat complicated array of genetic data sets that allow for a good characterization of the genomic structure of the "Red locus," which was previously mapped to a relatively narrow region on the Z-chromosome. They also present an analysis of gene expression in developing black versus red head feathers (albeit based on small sample size), and conduct a simulation study to test whether the observed data could have been generated by neutral processes.

On the first point of characterizing patterns of genetic diversity and divergence at the Red locus, the data are robust and yield an interesting result: the 'red' and 'black' alleles map to a ~72 kbp region of strongly elevated divergence and linkage disequilibrium with a relatively sharp transition to baseline levels on both ends. This pattern is what one might expect from an inversion, but long range PCR products spanning the edges of the ~72 kbp region reveal no evidence of an inversion. I highlight a few concerns about this aspect of the study below:

- 1) Difficulty keeping track of all the different data sets. I gather that this study represents a collaboration that developed from what were originally two or perhaps three independent research efforts. Whether that is an accurate inference or not, it is often difficult to follow which data set is being used for a given result/analysis. Likewise, it is difficult to keep track of the level of genomic coverage and quality (e.g., Sanger versus next gen and at what sequencing depth) that each data set provides. It also seems that results for some data sets are never presented (e.g., the WGS data and results for the M2 and M3 markers). Partial solutions to this problem might include: 1) a supplementary table that summarizes the different data sets and includes basic information about population sources, sample sizes, as well as number of loci, overall length of the sequences included, and/or genomic coverage and sequencing depth as appropriate, and 2) identification in each figure legend of the data set used, both in the main paper and the supplementary information doc.
- 2) Definition of the "Red locus" versus "core of the Red locus." It was not clear to me that "Red locus" was consistently defined throughout the paper and in all of the analyses presented. More importantly, I did not see any value in distinguishing between the locus and the core of the locus. I would strongly recommend that the "Red locus" be defined as being equivalent to the region of strongly elevated F_{st} and LD (the limits of which can be based on some objective criterion/criteria) and that adjacent sequences upstream and downstream of that region be referred to as "flanking regions." I suspect that the larger region included as part of the "Red locus" reflects either a previous, approximate estimate of its location based on the earlier mapping study, or operational decisions about regions to sequence and include in various analyses. What biological reality argues for including as part of the Red locus anything outside of the region of elevated F_{st} and LD?
- 3) The 9-locus microsatellite data set is not particularly impressive in comparison to the other data sets presented and provides relatively little power for detecting subtle patterns of population structure, if they exist. While the sampling for the RADSeq data is rather odd (i.e., 22 black males versus 8 red females and two red males), I would be interested to see what those data indicate about genome-wide or autosomal divergence between the color morphs (e.g., using the fineRADstructure software). This is a key question because previous papers on this system (e.g., Pryke & Griffith 2009 Evolution) seemed to assume some level of population differentiation between the morphs that would provide the basis/opportunity for genetic incompatibilities among loci (see below).

As noted, the gene expression data are based on small sample size, which prevents any statistically robust inferences about differences between the morphs. Moreover, the results section on gene expression transitions from identification of putative DE genes and pathways that might be involved in

feather development and/or coloration, to a summary of the genes follistatin more directly regulates based on other studies, to analyses of sequence conservation across the Red locus and follistatin coding region. I suspect that the authors are entirely correct in suggesting that the fates of feather precursors are determined earlier in development, but this section of the paper does not hold together particularly well and does relatively little to advance our understanding of how the red versus black alleles generate their myriad phenotypic effects, other than to rule out structural changes in the follistatin gene itself.

The third aim of the paper includes characterizing patterns of genetic diversity, divergence, and phylogenetic history across the Red locus and flanking regions, and then testing through the use of simulations whether entirely neutral processes could have generated the observed data. The authors take great pains to develop an unbiased approach to comparing observed and simulated data (e.g., structuring the simulation to match the empirical sample and assuming, in turn, that either the red allele or black allele is derived) but I still have some concerns about what exactly this analysis reveals. The point above about clearly defining the "Red locus" is potentially important not just for making the data and results easier for readers to understand, but also because it may influence the estimate of recombination rate used in the simulation study. Based on all of the data presented, either recombination is suppressed within "Red locus" (i.e., within the region of elevated LD), or recombinant alleles experience strong negative selection, or both. If the goal is to distinguish between these alternatives, it seems to me that estimating and simulating recombination across the Red locus and flanking regions requires careful consideration of both the empirical data and the simulations and an explicit rationale for defining the limits of the Red locus. Of particular concern is to what extent the empirical estimate of recombination (i.e., a minimum of 58 events in the sample) reflects recombination events that occurred in the "flanking regions" and/or recombination events that involved two alleles from the same lineage, i.e., two black alleles or two red alleles, respectively. One might ask: 1) is recombination generally reduced in the Red locus?; or 2) is it reduced only in red/black heterozygotes?; or 3) is there no reduction in physical recombination rate but instead the pattern of LD is generated by selection against recombinants"; or 4) some combination of the above? As I understand it, the simulation assumes both neutrality and a particular recombination rate that applies to the entire Red locus and flanking regions. The data clearly do not fit this model. While neutrality seems unlikely given the biology of the system, it's not clear to me that rejection of the null leads to a strong inference of balancing selection rather than a more general inference that the history and processes generating the empirical data are more complicated than the null model. What is the direct/positive evidence of balancing selection? Is it possible to test the different hypotheses about recombination noted above using multiple simulations with different assumptions?

Finally, I remain puzzled about the overall explanation for the origin, dynamics and evolutionary maintenance of color polymorphism in Gouldian finches, though it may not be fair to hold this paper and this set of authors entirely responsible for that. For example, Pryke & Griffith (2007, *J. Evol. Biol.*) and Pryke (2010) seem to tell somewhat different stories about mate preferences (do all females prefer red or is a preference for an individual's "own morph" genetically determined?), whereas Bolton et al. 2017 raise significant questions about whether wild birds are affected by the huge differences in survival and apparent genetic incompatibilities observed in captive birds. Within the context of the present paper and looking back at Pryke (2010), a clear hypothesis about which combinations of alleles at which loci are incompatible seems to be lacking. The present paper shows little or no genome-wide divergence between morphs. If so, then there would seem to be little or no opportunity for incompatibility between the Z-linked Red locus and autosomal loci; or at least, little or no opportunity for avoiding such incompatibilities through mate choice/assortative mating – i.e., if head color does not predict genotypes at other loci. Incompatibility might occur between the red and black alleles but presumably that would only be expressed in heterozygous males, whereas Pryke (2010) showed that females from mixed pairs suffered the lowest fitness, consistent with Haldane's rule. I

note that Pryke's (2010) analysis focused on offspring survival as a function of parental genotypes (and whether they matched or not) rather than offspring genotypes per se, making it somewhat difficult to interpret the results of that study. In the present study and given a lack of evidence for an inversion, the authors speculate that there is selection against recombinant alleles, implying that multiple elements within the red and black alleles, respectively, are co-adapted and must be maintained together in the same haplotype to avoid reduced fitness. It's not clear to me how that hypothesis would be reconciled with Pryke's results, but more generally, specific hypotheses about the nature of potential genetic incompatibilities in this system seem to be lacking.

All of this is potentially relevant to the question of what exactly can be inferred from the rejection of the null neutral model used in the authors' simulations. The present paper concludes that some form of balancing selection is in effect and discusses several alternative forms, but it seems possible that rejection of the null model might point to other differences between the simulated data and the empirical data (e.g., some mechanism other than an inversion that limits recombination).

I am returning the manuscript file with numerous questions, comments, and suggested edits embedded using track changes. Below, I will highlight several specific points that I think are important to address and will then provide some specific comments on the supplementary information document.

Comments keyed to line numbers:

99 and elsewhere: "core region" – Given the steep transition in F_{st} values and LD, is there really any useful distinction between "the region" and "the core region"? Moving through the rest of the manuscript, I found the term "core region" to be not particularly helpful and potentially confusing. If a larger region was identified in previous studies or with the RAD-seq data, why not go ahead and use the refined estimate provided here and consider the region of strongly elevated LD as "the Red locus" and the regions to the immediate left and right as flanking regions?

126-135: it's not entirely clear how this discussion of follistatin and the genes it regulates connects to the specific genes/pathways identified in the previous paragraph.

205-206: Isn't this result terribly problematic to the hypothesis that genetic incompatibilities are responsible for lower survival in the offspring of mixed-morph and mixed genotype pairs? There could be incompatibility between the Zr and ZR alleles at the red locus, but this should affect only the survival of heterozygous males. Why exactly do the offspring of mixed morph pairs suffer lower survival as a function of their parents' genotypes rather than as a function of their own genotypes? And, if the morphs are not differentiated genome-wide, what generates a higher likelihood of interlocus incompatibility in the female offspring of mixed morph pairs versus same morph pairs?

231: I am curious to know; 1) if recombination within the red locus is still detected if the red locus is defined as coinciding with the region of nearly maximal F_{st} and elevated LD, and 2) if the analysis can distinguish between recombination between alleles within one of the two divergent lineages versus recombination between divergent alleles. Put another way, how did recombination rate vary across the red locus? Is it just at the ends, beyond the "core"? And, does the analysis distinguish within lineage recombination from between lineage recombination? The strong LD at the red locus would seem to suggest that there is more than one selected/functional site across the locus. Otherwise, it's not clear how these patterns of divergence could be maintained without an inversion.

331: Herein lies the puzzle: what is the nature of these genetic incompatibilities assuming there is no population-level divergence between morphs? How does breeding between morphs generate

incompatibilities if no other loci in the genome are associated with the red locus?

579: This seems problematic. Given recombination and given that the samples were allocated to groups based on genotype at the Red locus, a comparison between the two groups at the reference loci is not necessarily going to measure the same thing (e.g., maximal historical divergence for the Red locus but not for the reference loci). To make the comparison more comparable, why not compare the two maximally divergent alleles at each reference locus?

Supplementary doc:

Line 86: gBGC – it's not clear how or if this is used in the following simulations/analyses.

Lines 145-146: seems likely that there would be at least some recurrent mutation over such a large region.

Lines 137-165: I have two concerns/questions here. First, for the empirical data, is it possible to distinguish between (and/or estimate the number of) recombination events that occurred between two red alleles or between two black alleles (i.e., how many within allelic lineage recombination events versus between allelic lineage recombination events)? Second, this section does not clearly define the Red locus, but the next section mentions simulating data for "a 99,669 bp region (equivalent in length to the Red locus)." If recombination in the empirical data was estimated for a comparable region, which includes "flanking regions" of lower F_{st} and LD, how many of the minimum of 58 recombination events occurred in the "flanking regions" rather than in the ~ 72 kbp region of elevated F_{st} and LD? Based on all of the data presented, either recombination is suppressed in the "Red locus" (i.e., within the region of elevated LD), or recombinant alleles experience strong negative selection, or both. Is the goal to distinguish between these alternatives? It seems to me that estimating and simulating recombination across the Red locus and flanking regions requires careful consideration of both the empirical data and the simulations and an explicit rationale for defining the limits of the Red locus.

170: what defines the limits of the "locus" - why not define the locus as the core region? Why not say, "we simulated a 99,669 bp region representing the red locus (xx,xxx bp) plus flanking regions"?

286: see questions above.

Suppl. Fig. 1: does this figure plot RADseq or WGS results? Based on the methods section, WGS data were collected for a sample of 24 captive birds, of which 17 were analyzed. If this is WGS data (and based on all the other results presented), there should be ~ 14 consecutive 5kb windows on the Z-chromosome with F_{st} close to 1. I don't see that in the figure. If these are RADseq results, where are the WGS results presented?

Suppl. Fig. 2: does this fig show all RAD loci in the ~ 5 Mbp region covered or only selected loci/SNPs? If the latter, how were they selected? What do the results for M2 and M3 look like? Are M2 and M3 also perfectly (or nearly perfectly) associated with phenotype?

Suppl. Fig. 4: does this gene have a single exon?

Suppl. Fig. 5: What does the top panel labeled "OtherRefSeq" illustrate? Rather than "pre-calculated," which is vague, how about "PhastCon values obtained from..." I think the red box should highlight the ~ 72 kbp region of clearly elevated F_{st} and LD and that this region should be defined as the "Red locus" (see other comments).

Suppl. Fig. 9: descriptions for e, f, g, and h in legend are not labeled correctly

Suppl. Fig. 15: is this tree based on complete coverage of the region from MiSeq data?

Suppl. Table 7: it would be helpful to highlight the two (one on each end?) fragments that span the transitions on left and right from high F_{st} and high LD to much lower values of each. It seems to me that it is only those two fragments that really matter in terms of refuting the hypothesis of an inversion.

Suppl. Table 9: what's the rationale for defining the Red locus as encompassing the entire ~100 kbp region? See comments above.

Reviewer #2:

Remarks to the Author:

There continues to be uncertainty and controversy regarding the relative importance of multilocus supergenes (typically associated with genomic regions of low recombination such as inversions) versus pleiotropic single locus "switches" in the evolution of color polymorphisms associated with correlated phenotypic traits. Lately, some supergene examples have been characterized and received a good deal of attention, notably in birds. In this MS Kim and colleagues present evidence for a roughly 72kb intergenic sequence, that likely plays a role in the regulation of the follistatin gene, as being the only consistent difference genomic between the red and black morphs of Australia's Gouldian finch, a species in which the morphs differ in morphology, behavior, and physiology as well as head color. This species and its polymorphism are famous and visually striking, but a bit enigmatic in that some of the results for more or less domesticated strains and laboratory populations seem not to hold up well in studies of wild birds. The authors additionally find no evidence of an inversion being associated with the region of high differentiation and the evidence against an inversion seems to me persuasive.

I can see no obvious or severe problems with the claims presented by the authors, and the MS is generally well written with few errors. One point that seems to me unclear is why balancing selection and an inversion are treated as strict alternatives. If a population is polymorphic for the presence or absence of an inversion, one must still explain the persistence of that polymorphism, and balancing selection could be the answer. Similarly, inversions may not be a perfect barrier to gene flow between homologous inverted and non-inverted regions of chromosomes. What they are really seeking to explain is strong linkage disequilibrium along a substantial length of chromosome, with the alternative forms associated with different complex phenotypes.

The main criticism one can make is that the story is somewhat incomplete. The authors were unable to characterize transcriptomic differences between the morphs that relate directly to the putative switch locus, thus the mode of action of the apparent switch has not been elucidated, and they have not attempted a manipulation of the locus in question, so its role is not fully established. The main question for the Editor then is whether the story as presented—which is clearly important and interesting—is of sufficiently broad interest and impact as to justify publication in Nature Communications.

Specific Comments:

39: comma after common

39: claims to be models or emerging models are regrettably common and often inadequately

substantiated; better to make a more specific claim for importance or novelty

46: delete "to"

84: consistent seems strong with $p=.11$... but not significantly different from expectation is reasonable

188: good to provide a citation here

254: better to explain a little more clearly how this scenario leads to frequency-dependence

263: run on sentence

267: sentence runs on some

280: family-wise?

293: a little more explanation of how to interpret c would be helpful

347: nice if this point could be fit into the main text, though clearly space is an issue

Suppl 301: not Kim at al

Suppl 448: why non-alphabetical legend organization? Correct?

Suppl 458: 10a?

Suppl 506: could mention lack of field evidence for assortative mating here

Reviewer #3:

Remarks to the Author:

This is an impressive and well carried out study examining the genetic basis of a color polymorphism in Gouldian finches using a variety of approaches. The authors demonstrate that the divergence between the red and the black morph reside to a 72 kb region on the Z chromosome, consistent with earlier findings of this polymorphism being located in this region on Z. There are several interesting findings: firstly, given the many differences in a plethora of traits between the morphs, that this is not due to an inversion as seen in many other species but rather (most likely) due to pleiotropic effect. Second, that it locates to a intergenic region suggesting a regulatory role (also well worth keeping in mind that 60-70% of all GWAS hits are intergenic) rather than a coding sequence change. Third, using coalcent simulations, the authors demonstrate that this polymorphism is likely maintained by balancing selection and arose in the Gouldian finch lineage.

Overall, this ms is very well written and carried out, although I note that I do not have expertise in all the analyses performed by the authors. It provides a nice contrast to the recent findings of the involvement of inversions ('supergenes') causing polymorphisms in several bird species (ruffs, zebra finches etc).

From my reading the mapping and identification of the region seems very solid and it is very interesting that the same gene seem to be involved in color divergence in a different species pair (golden and blue winged warblers). Given the finding of a TE within the Red locus I did wonder if this could be linked to the polymorphism as well, even if it arose before divergence between the red and black morph it could facilitate divergence. This is interesting since TE is linked to color patterning and polymorphism in both *Heliconius* and *Biston betularia*. It is also not clear how This is even more so given that the analysis to demonstrate balancing selection as the mechanism for the maintenance of this polymorphism require strong assortative mating, somethign which is not observed in natural populations. Should one not also expect accummulation of deleterious mutations within the region since recombination is essentially not taking place? This is something that you could test with the resequencing data?

Reviewer #1 (Remarks to the Author):

This is an interesting study on a fascinating bird species with a color polymorphism that is also associated with a suite of behavioral and physiological traits. The authors present a somewhat complicated array of genetic data sets that allow for a good characterization of the genomic structure of the "Red locus," which was previously mapped to a relatively narrow region on the Z-chromosome. They also present an analysis of gene expression in developing black versus red head feathers (albeit based on small sample size), and conduct a simulation study to test whether the observed data could have been generated by neutral processes.

We thank the reviewer for their generous and enthusiastic comments. The suggestions on how to improve the manuscript are helpful, and we have taken these on board. There were a substantial number of comments from Reviewer #1, so we have added a Table below to answer them as fully as possible.

On the first point of characterizing patterns of genetic diversity and divergence at the Red locus, the data are robust and yield an interesting result: the 'red' and 'black' alleles map to a ~72 kbp region of strongly elevated divergence and linkage disequilibrium with a relatively sharp transition to baseline levels on both ends. This pattern is what one might expect from an inversion, but long range PCR products spanning the edges of the ~72 kbp region reveal no evidence of an inversion. I highlight a few concerns about this aspect of the study below:

1) Difficulty keeping track of all the different data sets. I gather that this study represents a collaboration that developed from what were originally two or perhaps three independent research efforts. Whether that is an accurate inference or not, it is often difficult to follow which data set is being used for a given result/analysis. Likewise, it is difficult to keep track of the level of genomic coverage and quality (e.g., Sanger versus next gen and at what sequencing depth) that each data set provides. It also seems that results for some data sets are never presented (e.g., the WGS data and results for the M2 and M3 markers). Partial solutions to this problem might include: 1) a supplementary table that summarizes the different data sets and includes basic information about population sources, sample sizes, as well as number of loci, overall length of the sequences included, and/or genomic coverage and sequencing depth as appropriate, and 2) identification in each figure legend of the data set used, both in the main paper and the supplementary information doc.

Throughout the manuscript we have used only samples from a wild population except for the WGS, which used samples from a captive population. We have added the sources of data to each figure caption and added figure numbers in each section of the Methods to help readers to find the respective results more easily. We have clarified that the WGS data are shown in Supplementary Fig. 1 and added the genotyping results for the M2 and M3 markers to Supplementary Fig. 2. We have also clarified the definition of the Red locus and the core region in Method section "Genomic regions under consideration in the test for neutrality" and in Figures 1-2. We have removed an extra flanking region to reduce the confusion.

2) Definition of the "Red locus" versus "core of the Red locus." It was not clear to me that "Red locus" was consistently defined throughout the paper and in all of the analyses presented. More importantly, I did not see any value in distinguishing between the locus and the core of the locus. I would strongly recommend that the "Red locus" be defined as being equivalent to the region of strongly elevated F_{ST} and LD (the limits of which can be based on some objective criterion/criteria) and that adjacent sequences upstream and downstream of that region be referred to as "flanking regions." I suspect that the larger region included as part of the "Red locus" reflects either a previous, approximate estimate of its location based on the earlier mapping study, or operational decisions about regions to sequence and include in various analyses. What biological reality argues for including as part of the Red locus anything outside of the region of elevated F_{ST} and LD?

The definition of a 'locus' is often arbitrary, particularly when the causal site is not clearly identified. In a previous linkage mapping study (Kim et al 2016), the Red locus was mapped within a 7.2-cM (~8 Mb) interval, but the current study has refined it to an ~100kbp region including the ~72 kbp highly differentiated core regions.

In fact, we initially considered defining the Red locus as lying within the narrower interval, but it is difficult to clearly define the boundary between the core region and the remainder of the 100kbp region without identifying breakpoints (had it been within an inversion, which we have shown it is not) or being able to apply functional criteria. Although we have used $F_{ST} > 0.8$ to define the core region to highlight the rapid transition, and have now made this clear in the Methods and in Figs. 1-2, the choice of a specific value of F_{ST} is still arbitrary. While tightening the definition of the Red locus from 100kbp region to 72 kbp does not significantly improve the resolution, using the broader range is more conservative for now; our simulation results and HKA results would appear even more significant if we considered only the 72 kbp region of elevated F_{ST} .

However, we do agree that the reviewer's concern regarding the potential differences in recombination rate between the core and the flanking regions is relevant, so we now provide further information about the test we conducted before concluding that using the current definition in the subsequent sections is appropriate (see the response to the comments on lines 137-165, 145-146, and 231 of the original manuscript, below).

We agree that the extra "flanking region" upstream of (and not contiguous with) the Red locus does not add much information – it was not used in the simulations. Therefore, we have removed reference to this locus from the main text and Tables (Supplementary Table 4 & 6) accordingly, to reduce confusion.

** Kim, K. W., Griffith, S. C. & Burke, T. Linkage mapping of a polymorphic plumage locus associated with intermorph incompatibility in the Gouldian finch (*Erythrura gouldiae*). *Heredity* **116**, 409-416 (2016).*

3) The 9-locus microsatellite data set is not particularly impressive in comparison to the other data sets presented and provides relatively little power for detecting subtle patterns of population structure, if they exist. While the sampling for the RADSeq data is rather odd (i.e., 22 black males versus 8 red females and two red males), I would be interested to see what those data indicate about genome-wide or autosomal divergence between the color morphs (e.g., using the fineRADstructure software). This is a key question because previous papers on this system (e.g., Pryke & Griffith 2009 *Evolution*) seemed to assume some level of population differentiation between the morphs that would provide the basis/opportunity for genetic incompatibilities among loci (see below).

We used microsatellite data as this was an efficient way to test for population structure between the two morphs, given the reasonably large sample size. The initial sampling scheme for the RADSeq analysis (10 black males, 8 red females, and 2 red males) aimed to locate the Red locus more precisely, but again efficiently, given a restricted budget, rather than to make more detailed population genetic inference. At the Red locus, black males should be homozygous for the recessive (black) allele and red females should be hemizygous for the dominant allele. We did not know the genotypes of the two red males, which could be either heterozygous or homozygous for red. By treating red females as homozygous we were able to apply the dominance model in the GWAS analysis using plink, and so successfully detect the location of the Red locus. Twelve additional black male samples were available from other experiments and were subsequently included to increase the power further.

Although the Z chromosome sampling was not balanced for the population genetic inference, as the reviewer suggests we can indeed assume the autosomal chromosomes were randomly chosen with respect to Red, so that they can be used in the population genetic analysis. To confirm our STRUCTURE analysis, we have now performed the suggested fineRADstructure analysis using the autosomal RAD loci (Supplementary Fig. 6). Except for clusters caused by batch effects (amount of missing data per library), there was no population structure with regard to the head colour, consistent with the STRUCTURE analysis. As we further discuss below, and as the reviewer points out, testing for the genetic differentiation between morphs was an important indirect test for incompatibilities. Our results show that there is no widespread genomic differentiation between morphs.

As noted, the gene expression data are based on small sample size, which prevents any statistically robust inferences about differences between the morphs. Moreover, the results section on gene expression transitions from identification of putative DE genes and pathways that might be involved in feather development and/or coloration, to a summary of the genes follistatin more directly regulates based on other studies, to analyses of sequence conservation across the Red locus and follistatin coding region. I suspect that the authors are entirely correct in suggesting that the fates of feather precursors are determined earlier in development, but this section of the paper does not hold together particularly well and does relatively little to advance our understanding of how the red versus black alleles generate their myriad phenotypic effects, other than to rule out structural changes in the follistatin gene itself.

We agree that the gene expression data are of low power due to the small sample size but we provide the result as a suggestive list for future studies. However, it is interesting that multiple differentially expressed genes belong to a functionally correlated Gene Ontology network that has the potential to control feather development and pigmentation. We have now moved the description of the annotation network to the caption of Supplementary Figure 3, so reducing this aspect in the main text, and rephrased the main text to highlight the upstream position of FST in the regulatory network.

The third aim of the paper includes characterizing patterns of genetic diversity, divergence, and phylogenetic history across the Red locus and flanking regions, and then testing through the use of simulations whether entirely neutral processes could have generated the observed data. The authors take great pains to develop an unbiased approach to comparing observed and simulated data (e.g., structuring

the simulation to match the empirical sample and assuming, in turn, that either the red allele or black allele is derived) but I still have some concerns about what exactly this analysis reveals.

The point above about clearly defining the "Red locus" is potentially important not just for making the data and results easier for readers to understand, but also because it may influence the estimate of recombination rate used in the simulation study. Based on all of the data presented, either recombination is suppressed within "Red locus" (i.e., within the region of elevated LD), or recombinant alleles experience strong negative selection, or both. If the goal is to distinguish between these alternatives, it seems to me that estimating and simulating recombination across the Red locus and flanking regions requires careful consideration of both the empirical data and the simulations and an explicit rationale for defining the limits of the Red locus.

Of particular concern is to what extent the empirical estimate of recombination (i.e., a minimum of 58 events in the sample) reflects recombination events that occurred in the "flanking regions" and/or recombination events that involved two alleles from the same lineage, i.e., two black alleles or two red alleles, respectively. One might ask: 1) is recombination generally reduced in the Red locus?; or 2) is it reduced only in red/black heterozygotes?; or 3) is there no reduction in physical recombination rate but instead the pattern of LD is generated by selection against recombinants"; or 4) some combination of the above?

Regarding the various comments on the recombination rate, see the responses to the comments on lines 137-165, 145-146, and 231, below.

As I understand it, the simulation assumes both neutrality and a particular recombination rate that applies to the entire Red locus and flanking regions. The data clearly do not fit this model. While neutrality seems unlikely given the biology of the system, it's not clear to me that rejection of the null leads to a strong inference of balancing selection rather than a more general inference that the history and processes generating the empirical data are more complicated than the null model. What is the direct/positive evidence of balancing selection? Is it possible to test the different hypotheses about recombination noted above using multiple simulations with different assumptions?

Unfortunately, we cannot use the procedure we employed to simulate long-term balancing selection, because the methods that we used (SelSim, mbs) break down in this sort of scenario. However, rejecting the neutral model is the starting point in any attempt to look for evidence for

selection (Casillas and Barbadilla, 2017), and is the null model in widely used methods such as Tajima's D.

We are a little confused by the statement "the data clearly do not fit [the neutral] model", because the situation is complicated by the non-random sampling scheme at the Red locus. Our simulations have clearly demonstrated that, under this sampling scheme, genomic regions close to the site on which the sampling is based will have unusual polymorphism patterns (high nucleotide diversity, positive Tajima's D, etc. – see Figure 3 in the main text and Supplementary Figure 10), even under neutrality. Taken at face value, these patterns are consistent with those predicted by classical models of balancing selection. Our simulations serve to clearly indicate that the data are inconsistent with a model of neutral evolution and non-random sampling. It is on the basis of having rejected this null model that we set out to look for alternative explanations. Since we observe a large excess of pi, a highly positive Tajima's D, and unusually high divergence between alleles, over and above the effects of the sampling scheme, these are "positive evidence of balancing selection".

We also do not think that it is possible to test the different hypotheses about recombination that the reviewer suggests: first, we think that they are mostly a function of a local reduction in the recombination rate, but second, it is not possible to incorporate different rates of recombination for the different classes using the coalescent simulation machinery that we employ. We do deal more extensively with the reviewer's comments on recombination rate in our responses to the comments on lines 137-165, 145-146, and 231 of the original manuscript.

** Casillas, S., Barbadilla, A. Molecular population genetics. *Genetics* **205**, 1003–1035 (2017)*

Finally, I remain puzzled about the overall explanation for the origin, dynamics and evolutionary maintenance of color polymorphism in Gouldian finches, though it may not be fair to hold this paper and this set of authors entirely responsible for that. For example, Pryke & Griffith (2007, *J. Evol. Biol.*) and Pryke (2010) seem to tell somewhat different stories about mate preferences (do all females prefer red or is a preference for an individual's "own morph" genetically determined?), whereas Bolton et al. 2017 raise significant questions about whether wild birds are affected by the huge differences in survival and apparent genetic incompatibilities observed in captive birds. Within the context of the present paper and looking back at Pryke (2010), a clear hypothesis about which combinations of alleles at which loci are incompatible seems to be lacking. The present paper shows little or no genome-wide divergence between morphs. If so, then there would seem to be little or no opportunity for incompatibility between the Z-linked Red locus and autosomal loci; or at least, little or no opportunity for avoiding such

incompatibilities through mate choice/assortative mating – i.e., if head color does not predict genotypes at other loci. Incompatibility might occur between the red and black alleles but presumably that would only be expressed in heterozygous males, whereas Pryke (2010) showed that females from mixed pairs suffered the lowest fitness, consistent with Haldane's rule. I note that Pryke's (2010) analysis focused on offspring survival as a function of parental genotypes (and whether they matched or not) rather than offspring genotypes per se, making it somewhat difficult to interpret the results of that study. In the present study and given a lack of evidence for an inversion, the authors speculate that there is selection against recombinant alleles, implying that multiple elements within the red and black alleles, respectively, are co-adapted and must be maintained together in the same haplotype to avoid reduced fitness. It's not clear to me how that hypothesis would be reconciled with Pryke's results, but more generally, specific hypotheses about the nature of potential genetic incompatibilities in this system seem to be lacking.

Although the genetic basis of the incompatibilities observed in the captive population is beyond the scope of our study, one of the goals in this paper was to indirectly test if the incompatibilities described in captive populations are occurring in the wild population by examining the pattern of genomic differentiation. Clearly, our results including the test for HWE (Supplementary Table 3), the F_{ST} scan between morphs (Supplementary Figure 1), STRUCTURE analysis (Supplementary Figure 7), the pattern of LD around the Red locus (Figure 1), and the new fineRADstructure analysis (Supplementary Figure 6) show no or little evidence of any widespread incompatibility leading to genomic differentiation between the morphs.

We can only speculate that artificial selection and/or inbreeding in the captive population might have led to the discrepancy between the captive population studied by Pryke et al. and the wild population. First, it is noteworthy that LD in the captive population we studied here has significantly increased compared to that in the wild population. In our previous linkage mapping study (Kim et al, 2016) – from the captive population that was also the source of the birds used for the studies of incompatibility (Pryke et al. 2009) – the region of zero recombination with the causal site(s) for head colour extends up to ~8 Mb. Second, there may have been artificial selection by breeders on the captive birds, for example, to obtain brighter, purer head colours. Third, the effects reported by Pryke et al. might be exacerbated by the artificial environment used, and may not occur in nature. The lack of evidence for widespread incompatibilities (in terms of genome-wide population structure) in the wild population suggests that any effects of the Red locus alone may not be enough to promote the formation of incompatibilities elsewhere in the genome.

** Pryke, S. R. & Griffith, S. C. Postzygotic genetic incompatibility between sympatric color morphs. Evolution 63, 793-798 (2009).*

All of this is potentially relevant to the question of what exactly can be inferred from the rejection of the null neutral model used in the authors' simulations. The present paper concludes that some form of balancing selection is in effect and discusses several alternative forms, but it seems possible that rejection of the null model might point to other differences between the simulated data and the empirical data (e.g., some mechanism other than an inversion that limits recombination).

I am returning the manuscript file with numerous questions, comments, and suggested edits embedded using track changes. Below, I will highlight several specific points that I think are important to address and will then provide some specific comments on the supplementary information document.

Comments keyed to line numbers:

** The line numbers used by Reviewer #1 differed from our original manuscript (NCOMMS-18-10766-T - 18034016). We have inferred from the comments what was being referenced and have used the reviewer's line numbers as references in the response.*

99 and elsewhere: "core region" – Given the steep transition in F_{st} values and LD, is there really any useful distinction between "the region" and "the core region"? Moving through the rest of the manuscript, I found the term "core region" to be not particularly helpful and potentially confusing. If a larger region was identified in previous studies or with the RAD-seq data, why not go ahead and use the refined estimate provided here and consider the region of strongly elevated LD as "the Red locus" and the regions to the immediate left and right as flanking regions.

See above response.

126-135: it's not entirely clear how this discussion of follistatin and the genes it regulates connects to the specific genes/pathways identified in the previous paragraph.

We have now moved an entire section about the transcriptomic data to the caption of Supplementary Fig. 3, and briefly mention that there were some DE genes known to interact with

FST. We have rephrased the test to state that multiple functional differences may require expression differences for genes that act upstream of the regulatory network in an earlier developmental stage.

205-206: Isn't this result terribly problematic to the hypothesis that genetic incompatibilities are responsible for lower survival in the offspring of mixed-morph and mixed genotype pairs? There could be incompatibility between the Zr and ZR alleles at the red locus, but this should affect only the survival of heterozygous males. Why exactly do the offspring of mixed morph pairs suffer lower survival as a function of their parents' genotypes rather than as a function of their own genotypes? And, if the morphs are not differentiated genome-wide, what generates a higher likelihood of interlocus incompatibility in the female offspring of mixed morph pairs versus same morph pairs?

We agree with the reviewer that these results are somewhat unexpected, given the reports of red/black incompatibilities in previous studies. In this study, we made no assumptions about incompatibilities between morphs in the wild but tested if the incompatibilities observed in captivity had led to genome-wide differentiation between morphs in the wild. Therefore, we do not think the result is itself problematic. However, our results, along with Bolton et al. (2017) suggest a new hypothesis: that the discrepancies are due to differences between the wild and captive populations, as discussed above. We now address this point explicitly in the discussion.

** Bolton, P. E. et al. The colour of paternity: extra-pair paternity in the wild Gouldian finch does not appear to be driven by genetic incompatibility between morphs. J. Evolution. Biol. 30, 174-190 (2017).*

231: I am curious to know; 1) if recombination within the red locus is still detected if the red locus is defined as coinciding with the region of nearly maximal Fst and elevated LD, and 2) if the analysis can distinguish between recombination between alleles within one of the two divergent lineages versus recombination between divergent alleles. Put another way, how did recombination rate vary across the red locus? Is it just at the ends, beyond the "core"? And, does the analysis distinguish within lineage recombination from between lineage recombination? The strong LD at the red locus would seem to suggest that there is more than one selected/functional site across the locus. Otherwise, it's not clear how these patterns of divergence could be maintained without an inversion.

The reviewer raises two very salient points. Regarding point 1), this was in fact something we investigated previously, and the short answer is, yes: if we carry out the R_M test across the 13,000-82,000 bp region of the Red locus (which corresponds to the block of elevated LD and F_{ST}) then we detect recombination events both within the red alleles ($R_M = 4$), the black alleles ($R_M = 15$), and in the combined sample ($R_M = 41$). The rate of R_M per bp is very similar (in fact slightly higher) for the block of high LD ($41 / 69,001 = 5.94e-4$), compared to the whole region ($58 / 99,699 = 5.82e-4$). We recognise that this is a legitimate concern and present these results in Supplementary Table 13 and discuss them in the Supplementary Text, section 'Estimating ρ , the population recombination rate'.

Regarding point 2), whether the analysis can distinguish between inter- vs. intra-class recombination events, see the response to Referee 1's comment on Lines 137-165 of the original manuscript, below. We would also like to add that we are not trying to make inferences about parameters to do with recombination and selection in the way that the reviewer suggests – rather, we have tried to construct a null model against which our hypothesis of balancing selection can be tested. Our rejection of the null hypothesis could lead to future analyses that try to parameterise some of the processes that are of interest (the nature of recombination and selection on specific sites) but (a) we are not aware of any method that currently allows us to simulate more complicated situations than the one we have here, while still incorporating our sampling scheme – the coalescent simulators we use are not capable of modelling more than one site under selection – and (b) the rejection of the null hypothesis is a necessary first step towards this end.

We agree that, in the absence of an inversion, the block of LD does suggest the presence of more than one functional/selected site. However, as we state above, our simulations are designed to replicate the null hypothesis of neutrality and a sampling scheme conditioned on a single site, so including two selected sites is unnecessary. Any inference about multiple selected sites also relies on the rejection of our null model as a first step.

331: Herein lies the puzzle: what is the nature of these genetic incompatibilities assuming there is no population-level divergence between morphs? How does breeding between morphs generate incompatibilities if no other loci in the genome are associated with the red locus?

See above response.

579: This seems problematic. Given recombination and given that the samples were allocated to groups based on genotype at the Red locus, a comparison between the two groups at the reference loci is not

necessarily going to measure the same thing (e.g., maximal historical divergence for the Red locus but not for the reference loci). To make the comparison more comparable, why not compare the two maximally divergent alleles at each reference locus?

We think that the test suggested by the reviewer is likely to be extremely conservative, because it is inconsistent with how our data were generated (our sampling was based on the alleles' association with morphs only, without any regard to divergence or any sequence-based statistics). However, if we carry out the test that the reviewer suggests with D_{xy} (Panel A below) or π (Panel B) (it is not possible to calculate Tajima's D or F_{st} with a sample of two sequences), the Red locus still exhibits patterns of polymorphism that exceed the 95% confidence intervals of the values at the reference loci. This excess of diversity and divergence is consistent with balancing selection.

Supplementary doc:

Line 86: gBGC – it's not clear how or if this is used in the following simulations/analyses.

gBGC was not incorporated in any further analyses beyond allowing for it when we estimated the demographic history of this population. We have included a statement to this effect at the end of the Supplementary Text section “Testing for a change in population size and gBGC”.

Lines 145-146: seems likely that there would be at least some recurrent mutation over such a large region.

This may be true, but it is less likely that it has a major effect. First, our R_M analysis using within-morph data alone, which has a very low diversity level (and hence negligible chance for homoplasy), still reveals evidence for recombination (new Supplementary Table 13). Second, our conclusions are robust to the recombination rate that we infer and simulate. Below, we present analogous results to those in Figure 3 in the main text, but with $\rho = 0.0002$ as opposed to $\rho = 0.004$, which is a reduction of more than an order of magnitude. A reduction compared to the estimated value of ρ is what we expect if some of the recombination events we infer using R_M were due to recurrent mutation. As an aside, 0.0002 is approximately the value of ρ inferred by LDhat, both within the red and within the black allelic classes at the Red locus. LDhat allows recurrent mutation. Please bear in mind, though, that the strong LD between the allelic classes prevented us from using LDhat to assay recombination in the combined sample at the Red locus, as we originally stated in the supplementary information.

As is apparent from the figures below, the observed polymorphism at the Red locus lies outside the 95% CIs from the simulations, even with this reduced recombination rate. This is true for a derived allele frequency of 0.144 (panels A-D) and a DAF of 0.856 (panels E-H).

Lines 137-165: I have two concerns/questions here. First, for the empirical data, is it possible to distinguish between (and/or estimate the number of) recombination events that occurred between two red alleles or between two black alleles (i.e., how many within allelic lineage recombination events versus between allelic lineage recombination events)?

See also the response (above) to comment on line 231 of the original manuscript. We detected recombination events within the red and within the black alleles for the whole region and for the region of elevated LD, and a greater number of recombination events when the samples are combined, which implies the presence of recombination events between allelic lineages. Again, we think this is a worthwhile comment and we have presented these results in Supplementary Table 13 and mention them in the Supplementary Text.

Second, this section does not clearly define the Red locus, but the next section mentions simulating data for "a 99,669 bp region (equivalent in length to the Red locus)." If recombination in the empirical data was estimated for a comparable region, which includes "flanking regions" of lower F_{st} and LD, how many of the minimum of 58 recombination events occurred in the "flanking regions" rather than in the ~72 kbp region of elevated F_{st} and LD? Based on all of the data presented, either recombination is suppressed in the "Red locus" (i.e., within the region of elevated LD), or recombinant alleles experience strong negative selection, or both. Is the goal to distinguish between these alternatives? It seems to me that estimating and simulating recombination across the Red locus and flanking regions requires careful

consideration of both the empirical data and the simulations and an explicit rationale for defining the limits of the Red locus.

Regarding recombination across different parts of the region under consideration, see the response to the comment above as well as the response to Referee 1's the comment on line 231 of the original manuscript. These suggest that our treatment of recombination at the Red locus for the purposes of testing our null model was conservative.

We do not claim to be able to distinguish between alternative explanations for an effective or real suppression of recombination across this region - and we do not think that this is possible with population genetic data alone. Our goal is to incorporate recombination into our analyses to a best approximation in an effort to test our null hypothesis of neutral evolution, not to conduct model choice between the two alternatives that the reviewer presents. We would also like to reiterate the robustness of our results to a recombination rate even an order of magnitude below the one that we infer using R_m - see our response to Referee 1's comment on lines 145-156, above.

170: what defines the limits of the "locus" - why not define the locus as the core region? Why not say, "we simulated a 99,669 bp region representing the red locus (xx,xxx bp) plus flanking regions"?

See above response for the definition of the Red locus.

286: see questions above.

See above response for the definition of the Red locus.

Suppl. Fig. 1: does this figure plot RADseq or WGS results? Based on the methods section, WGS data were collected for a sample of 24 captive birds, of which 17 were analyzed. If this is WGS data (and based on all the other results presented), there should be ~14 consecutive 5kbp windows on the Z-chromosome with F_{st} close to 1. I don't see that in the figure. If these are RADseq results, where are the WGS results presented?

This was a result from WGS using a sliding-window approach. Unfortunately, the nearest available windows flanking the single window (5kbp from 46,580,001 bp) on the Red locus were located at 310 kbp and 410 kbp away from the focal window, respectively.

We now present site-by-site F_{ST} from both RADSeq (wild population) and WGS (captive population), and the frequency distribution of F_{ST} values in Supplementary Fig. 1. The sources of the data are provided in the figure caption. After filtering, there were only two SNPs within the core of the Red locus for each (RADSeq and WGS) dataset, so the pattern discovered in Fig. 2 could not be obtained using these data.

Suppl. Fig. 2: does this fig show all RAD loci in the ~5Mbp region covered or only selected loci/SNPs? If the latter, how were they selected? What do the results for M2 and M3 look like? Are M2 and M3 also perfectly (or nearly perfectly) associated with phenotype?

This figure shows all RAD loci in the ~5 Mbp interval around the Red locus. Due to the density of the RAD loci there is only one RADSeq locus (with two SNPs) that overlaps with the Red locus. We have now included the genotype results for M1~M3 in Supplementary Fig. 2. Two recombination events between black- and red-linked alleles are visible in the genotypes of red males. A single black female that has red-linked alleles for all three markers is likely to represent a phenotyping error due to the less bright red head colour of females in the wild population.

Suppl. Fig. 4: does this gene have a single exon?

Six exonic sequences for zebra finch were obtained from Ensembl (<https://www.ensembl.org/>). More details have been added to the caption of Supplementary Figure 4.

Suppl. Fig. 5: What does the top panel labeled "OtherRefSeq" illustrate? Rather than "pre-calculated," which is vague, how about "PhastCon values obtained from..." I think the red box should highlight the ~72 kbp region of clearly elevated F_{st} and LD and that this region should be defined as the "Red locus" (see other comments).

We have now added descriptions for the top panel and the source of PhastCon values, as suggested. We have removed the panel "OtherRefSeq", which shows gene models for other species to reduce confusion. We have now added the position of the 'core' of the Red locus in the top panel.

Suppl. Fig. 9: descriptions for e, f, g, and h in legend are not labeled correctly

These are now correctly labelled

Suppl. Fig. 15: is this tree based on complete coverage of the region from MiSeq data?

This is based on the entire core region from MiSeq data.

Suppl. Table 7: it would be helpful to highlight the two (one on each end?) fragments that span the transitions on left and right from high F_{st} and high LD to much lower values of each. It seems to me that it is only those two fragments that really matter in terms of refuting the hypothesis of an inversion.

Approximate position of the 'core' of the Red locus is now shaded.

Suppl. Table 9: what's the rationale for defining the Red locus as encompassing the entire ~100 kbp region? See comments above.

See above response for the definition of the Red locus.

Table 1. Response to Reviewer 1. * Line numbers are based on "Reviewer #1 Review Attachment #1.pdf".

No.	Line*	Original	Comment / suggestion	Revision
1	38	Instead	Comment [A1]: Balancing selection is not necessarily an alternative to an inversion. Both could be relevant in the same system.	We agree with the suggestion that on inversion and balancing selection should not be presented as alternatives, and have amended the abstract accordingly.
2	60	incompatibilities	Comment [A2]: Does prezygotic incompatibility simply take the form of assortative mating? And how does assortative mating result if all females prefer red males?	In captive populations, prezygotic incompatibility was observed in the form of assortative mating. Currently, there is no strong evidence of assortative mating in the wild population. We have made this clear by adding "but these were not fully tested in the wild population".
3	79	near-perfect	Comment [A3]: A single "CA" female scored as black? Is that the only exception? If so, might be good to say something like "with one exception"	Accepted. Added "with one exception in a female".
4	80	Supplementary Table 2, Supplementary Fig. 2;	Comment [A4]: Here and elsewhere: it may be less distracting to the reader to save supplementary methods information for the methods section or perhaps refer to it in the relevant figure legends.	Deleted "Supplementary Table 2, Supplementary Fig. 2;"
5	82	The genotypic frequencies in males did not depart from Hardy–Weinberg Equilibrium (Comment [A5]: This seems like a critical point in relation to inferences about assortative mating – is it really assortative in the wild population?	There is no strong evidence of assortative mating in the wild population and our results do not support it. This is now clarified in the introduction and also mentioned in the discussion.
6	84	for a sex-linked locus under random mating	Comment [A6]: What about allele frequency?	It is already stated that there is no departure from HWE, meaning that HWE is as expected given the allele frequency.
7	84	~1%	Comment [A7]: Isn't the HW expectation for homozygous males ~2% based on the estimated allele frequencies?	1 % refers to a prediction by Southern (1945) based on phenotypic frequency. We have changed this to "The low frequency of homozygous red males was not significantly different from expectation for a sex-linked locus under random mating (Supplementary Table 3) and a prediction based on phenotypic frequency".

8	95	the “core” region	Comment [A8]: Given the steep transition in Fst values and LD, is there really any useful distinction between the “region” and the “core region”? Moving through the rest of the manuscript, I found the “core region” distinction to be not particularly helpful and potentially confusing. If a larger region was delineated in previous studies, why not go ahead and use the refined estimate provided here and consider the region of strongly elevated LD as “the region” and sequence to the immediate left and right as flanking regions?	Throughout the manuscript the “ Red locus” is the full ~100 kbp sequenced region using MiSeq that contains the ~72 kbp “core region” that shows high divergence. Because we did not find any breakpoints or causal sites that distinguish the core and the regions adjacent to the stretch of high LD, we have left the definition of Red as the entire stretch of contiguous sequence obtained from the MiSeq. Because we did not find differences in recombination rates between this entire region and the ‘core’, we use parameters estimated using the entire region in the simulation. We believe this is a more conservative approach than using only the highly divergent region. See the main responses.
9	99	using long-range PCR	Comment [A9]: This long range PCR effort is impressive but it does not extend very far on either end of the region of elevated divergence, and the primer pair on the right end is considered part of the “locus” and is only a couple thousand base pairs beyond the “core” of the locus – assuming that the pattern of LD was known first, it seems like a better design would have been to focus on the ends rather than the middle. In any case, redefining “the region” or “the locus” as suggested in the previous comment may help to show that the long range PCR products effectively covered the ends of the locus.	See response to Comment [A8]
10	102	Supplementary Tables 6-7	Comment [A10]: Would be less distracting to leave methods information to the methods section or figure legends	Deleted "Supplementary Tables 6". Supplementary Tables 7 is the results of long-range PCR.
11	111	BMPR1B)	Comment [A11]: I don't see this in gene labelled in the supplementary figure	It is a DE gene in Supplementary Data Set 1. We have deleted BMPR1B to reduce confusion. Whole section was moved to Supplementary Fig. 3.
12	112	hair	Comment [A12]: Birds don't have hair.	Many genes are common to the development of feathers and hair.

13	116	to diverged feather developments	Comment [A13]: This seems a bit vague.	Rephrased to "differentially accumulated pigments might also be important in the development of different types of feather". Moved to Supplementary Fig. 3.
14	119	the pattern of DE genes suggests	Comment [A14]: Spell out DE – there are more than enough acronyms elsewhere in the paper!	accepted
15	119	the pattern of DE genes suggests	Comment [A15]: It's not clear to me HOW the observed patterns of DE genes lead to the conclusion expressed in this sentence.	Rephrased to "The multiple functional differences in gene expression suggest that the fates of feather precursors are determined by gene(s) that act upstream of the regulatory network at an earlier developmental stage".
16	148	Supplementary Tables 8	Comment [A16]: Table 8 is methods	Deleted "Supplementary Table 8" from text.
17	150	Supplementary Tables 8	Comment [A17]: Table 8 is methods	Deleted "Supplementary Table 8" from text.
18	152	as is the level of differentiation between ZR and Zr alleles	Comment [A18]: I don't think this is a fair comparison because there could be divergent lineages at other loci that do not correspond to morph – a different approach would be to compare divergence between ZR and Zr alleles at the red locus with maximal divergence of alleles at other loci	This point has been responded to more thoroughly in our main response to the reviewers' comments. We have carried out the test suggested by the reviewer and demonstrate that the Red locus is still an outlier. We also think that the suggested test is likely to be conservative, because it does not represent how the data were generated - sampling was carried out with respect to morph, not with respect to the most divergent alleles (no information from sequence-based statistics contributed to the design of the sampling scheme).
19	153	the core of the Red locus	Comment [A19]: As noted above the "core" concept does not seem particularly helpful.	See response to Comment [A8]
20	160	background	Comment [A20]: Shouldn't this be alleles or haplotypes?	We have changed "backgrounds" to "alleles".

21	161	between morphs	Comment [A21]: See comment above – this strikes me as two different kinds of comparisons. I think it would be better to identify the most divergent alleles at the reference loci – and ask about how max dxy compares to the red locus. Otherwise, it's not a fair comparison.	Please see the response to comment [A18] above.
22	163	between zebra finch and each morph	Comment [A22]: I think it would be better to talk about the haplotypes or alleles rather than the morphs. Or at a minimum, remind the reader that this analysis is based on females, such that morph is equivalent to allele at the red locus.	We have changed the text to refer to 'alleles', as per the reviewer's suggestion throughout the section entitled "The pattern of nucleotide diversity and divergence at the Red locus".
23	167	backgrounds	Comment [A23]: Again, shouldn't this be alleles or haplotypes?	Amended instances of 'background' to refer to 'alleles'.
24	170	the standard neutral model (Pcoalescent = 0.002)	Comment [A24]: I assume this is based on simulation, but that doesn't come through very well with just the phrase "standard neutral model"	Changed in the text to add brief methods: "using simulations in DNAsp v5"
25	172	pre- and postzygotic incompatibilities	Comment [A25]: What is the nature of the pre-zygotic incompatibilities?	See Supplementary Table 1. The pre-zygotic incompatibility is in the form of assortative mating in the captive population but no strong evidence of it is observed in the wild population.
26	178	a large stretch of the core region	Comment [A26]: As suggested above, I think it would be simpler to define the extent of the "red locus" as being equivalent to this region of nearly maximal F_{st} and strongly elevated LD. Note also that if there is some within lineage diversity, then F_{st} will be less than 1 even if there has been no recombination between the divergent alleles in this region of maximal divergence.	See response to Comment [A8]
27	179	the differentiation between morphs	Comment [A27]: Again, it might be helpful to remind the reader that this analysis is based on females.	Added "the differentiation between each morph's alleles"

28	181	unlinked autosomal microsatellite markers	Comment [A28]: Why not use the RAD data in PCA, STRUCTURE or fineRADstructure, for example? Nine microsatellites seems like a rather puny data set in the context of this study and not a particularly powerful means of detecting subtle differentiation (if it exists). Also, why not look for other outliers in both the RAD data and the low-depth genome-wide data?	We used microsatellite data for all available samples. Now we have added the fineRADstructure analysis using the RADSeq data.
29	182	The most-likely number of subpopulations was one (K = 1;	Comment [A29]: Is this really the best test available of possible genome-wide differentiation between morphs?	We used microsatellite data for all available samples. Now we have added the fineRADstructure analysis using the RADSeq data.
30	183	demonstrating a lack of differentiation with regard to colour morph across the genome outside of the Red locus in the wild	Comment [A30]: Isn't this result terribly problematic to the hypothesis that genetic incompatibilities are responsible for lower survival in the offspring of mixed-morph and mixed genotype pairs? There could be incompatibility between the Zr and ZR alleles at the red locus, but this should affect only the survival of heterozygous males. Why exactly do the offspring of mixed morph pairs suffer lower survival as a function of their parents genotypes rather than as a function of their own genotypes? And, if the mophs are not differentiated genome-wide, what generates a higher likelihood of interlocus incompatibility in the female offspring of mixed morph pairs versus same morph pairs?	Answered in main response. In short, we did not make the assumption that there should be incompatibilities in the wild population. We examined if incompatibilities described in the captive population are occurring in the wild population and found no support for the incompatibilities in the wild population. We think the discrepancy is caused by the differences between wild and captive population (see main response).

31	207	recombination within the Red locus	Comment [A31]: I am curious to know; 1) if recombination within the red locus is still detected if the red locus is defined as coinciding with the region of nearly maximal Fst and elevated LD, and 2) if the analysis can distinguish between recombination between alleles within one of the two divergent lineages versus recombination between divergent alleles. Put another way, how did recombination rate vary across the red locus? Is it just at the ends, beyond the “core”? And, does the analysis distinguish within lineage recombination from between lineage recombination? The strong LD at the red locus would seem to suggest that there is more than one selected/functional site across the locus. Otherwise, it’s not clear how these patterns of divergence could be maintained without an inversion.	We added new Supplementary Table 13 showing the recombination in the region encompassing the Red locus and compare the entire Red locus and the core region. There are negligible differences between per base recombination rate (In fact, the rate in the core region is slightly higher). Because we don't know the exact location of the causal site(s) and the proportion of the Red locus that is outside the region of elevated LD is low, we define the entire sequenced region as the Red locus. We also consider the multiple selected sites as a parsimonious explanation but we have chosen not to emphasise this because we don't yet know the causal site(s).
32	209	selection against recombinants	Comment [A32]: This also seems to imply that there are multiple functionally relevant sites across the red locus and that recombination of these sites generates incompatibilities.	We do consider multiple functional sites as a parsimonious explanation for the observed pattern but without knowing the exact sites we don't make a significant claim here.
33	211	the core of the Red locus	Comment [A33]: Seems like a different definition of “core” here. Again, it would be better to simplify by equating the red locus to the “core region” and referring to adjacent sequences as flanking regions.	Throughout the manuscript the Red locus is the full 100 kbp sequenced region that contains the ~72 kbp core region with high divergence. The "core" in the respective sentence has the same meaning.
34	219	Supplementary Fig. 13	Comment [A34]: Doesn't seem particularly relevant to the result being reported	It provides a comparison between loci in the core and outside of the core. Locus Ego176 also shows an example of highly significant differences in the Tajima's relative rate test.

35	231	the 3' and 5' LTRs in each haplotype	Comment [A35]: How many repeats are there in each locus? Same number/length in both black and red?	There is one each in the 3' and 5' ends of this region, one of which is likely to be a replicate produced during insertion of the element into the genome.
36	235	In a relative rate test	Comment [A36]: Of the entire locus or just the M1 region?	In the same sentence and Methods, we explained that this is done across the core of the Red locus.
37	246	ecologically sympatric	Comment [A37]: What does “ecologically sympatric” mean as compared to “sympatric”? Would “co-distributed taxa” be a better term?	co-distributed taxa
38	263	there is no evidence of assortative mating in wild populations	Comment [A38]: How is this observation and the other results of Bolton et al. reconciled with previous hypotheses and observations of genetic incompatibilities (e.g., Pryke 2010 Evolution)?	Answered in the main response. We think the discrepancy is caused by the differences between wild and captive population (see main response).
39	280	and genetic incompatibilities between morphs	Comment [A39]: Herein lies the puzzle: what is the nature of these genetic incompatibilities assuming there is no population-level divergence between morphs? How does breeding between morphs generate incompatibilities if no other loci in the genome are associated with the red locus?	This comment has been addressed in the main responses.
40	289	for the core region	Comment [A40]: Is the core region only 2 SNPs?	Two SNPs in the core region are used in the test.
41	292	Supplementary Fig. 1	Comment [A41]: 2?	accepted
42	297	three vertical red arrows indicate the locations of markers that were used to genotype the wild birds in association tests	Comment [A42]: Are results for M2 and M3 reported anywhere?	We have now added the results of M2 and M3 in Supplementary Fig. 2. Two recombinations between black and red alleles are visible in the red males' genotype. A single black female that has a red-linked allele is likely to represent a phenotyping error due to the less bright red head colour of females in the wild population.
43	298	association tests	Comment [A43]: Association with what?	between phenotype and genotype

44	300	using Sanger sequencing	Comment [A44]: How much data? What was the level of coverage across this region?	These are twenty-six short fragments (~1 kbp each) across the Red locus. The distribution of all SNPs found in these fragments is shown in the middle panel of Figure 1c. Sequence data that cover the entire region of the Red locus were obtained using the MiSeq data. This is now explained in the figure caption and Methods.
45	311	24 intronic reference loci	Comment [A45]: Same 6 + 6 female sample size?	This information is included at the end of the section entitled “Genomic regions under consideration in the test for neutrality”.
46	311	Z chromosome	Comment [A46]: what’s the window size? Needs explanation of colors in a-d – description for panel b uses labels only visible in panel f Suggestion: zoom in on a smaller y-axis range in e and g	Sliding-window analysis (window size of 1,000bp with a step of 500 bp) was used to obtain estimates. We have added descriptions of line colours. The scales were chosen to enable the comparison with the left panel. The figure caption has been amended accordingly.
47	317	recombination rate	Comment [A47]: Was a single, constant recombination rate assumed for all sites across the region?	Yes - we have tried to make this more explicit in the Supplementary Text section ‘Estimating ρ , the population recombination rate’.
48	318	observed data	Comment [A48]: For what data set? N= 6 + 6 females?	Yes, as the reviewer suggests. We have edited the legend for Figure 3 to make this clearer.
49	320	a	Comment [A49]: Why not zoom in on y-axis?	The scale was chosen to provide visual comparison with following d_{XY} plot.
50	332	the core of the Red locus	Comment [A50]: How defined?	See caption for Fig. 2d and Methods section ‘Genomic regions under consideration in the test for neutrality’.
51	337	c, pairwise genetic distance matrix	Comment [A51]: Not particularly instructive. The tree is sufficient to illustrate essentially the same thing	Deleted

52	345	A subset of samples was used for RADSeq and subsequent sequencing and genotyping. All 161 samples were used for the association test and a population structure analysis	Comment [A52]: Switch order of these two sentences	accepted
53	348	and confirmed	Comment [A53]: For all birds?	Yes, we genotyped all birds with msat markers that were used in previous linkage mapping (Kim et al 2016) as well. We have now deleted this sentence because the test using Z-002D and Z37B is enough for sexing.
54	350	For whole-genome sequencing (WGS),	Comment [A54]: I don't recall WGS data being reported above, although I note a single mention of WGS data at the beginning of the results	See Supplementary Figure 1. We have now included "WGS" to indicate the data source.
55	361	12 black males	Comment [A55]: What was the rationale for doing 22 black males, 8 red females and 2 red males? Seems a bit odd.	Added "This sampling scheme was to identify the Red locus efficiently with limited sample size in GWAS". See GWAS section which has now moved to be next to the RADSeq section.
56	369	the distribution of SNP depth per site (> 2),	Comment [A56]: What does this mean? Average of > 2 reads per individual? Greater than 2 reads for all individuals? How does this interact with the 70% per individual SNP calling rate? Were the two runs reasonably consistent in recovery of loci? What was average depth per locus per individual? Why is nothing more done with these data other than the plot in Fig 1a? E.g., PCA or STRUCTURE analysis of the RAD-seq data would be substantially more powerful for the analysis of possible population structure than 9 microsatellites.	We meant minimum depth for each site per individual (> 2) and maximum missing genotypes over all individuals ($< 30\%$). HiSeq2000 runs had more coverage than GAIIX runs. However, by applying a filter based on missing genotypes, we could effectively remove sites that were found only in the HiSeq2000 run. The mean depth per locus per individual differed between libraries (GAIIX: L1 = 3.2, L2 = 7.8; HiSeq2000: L3 = 13.2). However, because we mixed equal number of black and red birds in L1 and L2, any bias due to the coverage in the inference of structure between morphs will be limited. We have rephrased this section now. Originally, we did not design the RADSeq for population genetic analysis. We wanted to find the

				causative region on the Z chromosome at the lowest cost. We agree that the RADSeq data can be used for further analysis so we have now added the suggested fineRADstructure analysis.
57	378	This resulted in a mean of 18.7 million reads per individual (range 11.4–26.9 million reads)	Comment [A57]: 18.7M pairs of reads or 18.7M total reads? 18.7M x 150bp would be ~2.3x coverage for a 1.2B genome, or 4.6x if pairs of reads	We thank the reviewer for pointing this out. We were not careful in distinguishing paired and collapsed reads. We have now been explicit and reported the total number of reads to avoid confusion: Average 33.7 million total reads, which is approximately 3.6x coverage.
58	389	used a minor allele frequency (MAF) filter of 5%	Comment [A58]: This is not a good approach for possible analyses that rely on the allele frequency distribution or rare alleles specifically – e.g., recent co-ancestry can be assessed using rare alleles (e.g., using fineRADstructure for RAD data)	We acknowledge that this cut off is not ideal, but when working with the low coverage data that we generated it was prudent to be cautious. Our WGS data were from a captive population, so are not suitable for the population genetic analyses to detect population structure etc. This dataset was used only in detecting outlier loci via F_{ST} to identify and confirm the location of the Red locus.
59	393	known to be homozygous	Comment [A59]: or hemizygous females?	We have rephrased this part as “To properly correct for the ploidy of genotypes identified on the Z chromosome, we used the vcf-fix-ploidy function of VCFtools to assign one allele at each site for Z-linked loci in females. To increase the chance of detecting the location of the Red locus in an F_{ST} outlier analysis (Supplementary Fig. 1), we only included data from 17 individuals whose genotypes were known to be homozygous or hemizygous for alternative alleles at the Red locus based on the pedigree (black: $n_{\text{black male}} = 4$,

				$n_{\text{black female}} = 3$; red: $n_{\text{red female}} = 5$ and $n_{\text{yellow female}} = 5$)”.
60	396	mean F_{ST}	Comment [A60]: For the Z-chromosome, was this calculated with appropriate consideration of ploidy – i.e., total of 11 black chromosomes and 10 red chromosomes?	We used the vcf-fix-ploidy function of VCFtools to assign one allele at each site for the genotypes of hemizygous Z-linked loci in females. We obtained the allele frequencies based on the number of chromosomes we sampled (e.g. total of 11 and 10 alleles for black and red birds respectively for Z-chromosome) and used these in the calculation of F_{ST} (see Methods section ‘Estimates of DNA polymorphism and divergence’ for calculation of F_{ST}). We now present site-by-site F_{ST} scans for wild and captive populations in Supplementary Fig. 1. Text was amended accordingly.
61	397	to identify highly differentiated genomic windows	Comment [A61]: Are any of the WGS results reported above? If so, it’s not clear how they were used.	See F_{ST} scans for wild and captive populations using RADSeq and WGS data, respectively, in Supplementary Figure 1.
62	401	genome-wide association study (GWAS) in 32 RAD-sequenced samples	Comment [A62]: Are these results illustrated either in the main manuscript or in the supplementary information? The results section above says WGS and RAD-seq data were combined to identify the Z-linked region, but it’s not really clear how. Also not clearly stated or shown that NO other portions of the genome were associated with head color.	We meant that we used both RADSeq and WGS, but did not combine these data. The RADSeq result is presented in Fig. 1 and WGS data are shown in Supplementary Figure 1. In "identification of the Red locus" section, we deleted "a combination of ", for clarity. In the GWAS result (Fig. 1) there is only one significant point within the candidate interval from linkage mapping that passed the permutation test.
63	420	M2 and M3, Supplementary Table 2	Comment [A63]: Are results for these loci shown in main text or supplementary? What do these loci tell us?	We have now added the results of M2 and M3 in Supplementary Fig. 2. Two recombination between black and red alleles are visible in the red males’ genotype. A single black female that has red-linked allele is likely to represent a phenotyping error due to the less bright red head colour of females in the wild population.

64	425	twenty-six short fragments in the upstream region	Comment [A64]: What fraction of the region was covered by sequence data?	These are twenty-six short fragments (~1kbp each) across the Red locus. The distribution of all SNPs found in these fragments is shown in the middle panel of Figure 1c. Sequence data that cover the entire region of the Red locus were obtained using the MiSeq.
65	428	16 females ($n_{black} = 8$, $n_{red} = 8$)	Comment [A65]: From what population/source?	Throughout the manuscript we use only samples from a single wild population, except for the WGS data. This is mentioned in "DNA samples" section of Methods.
66	432	An association test for these individuals was performed	Comment [A66]: As described above?	It was performed using Haploview. We have rephrased to clarify as "Association tests between these haplotype SNPs and phenotype, and the estimation of pairwise LD between loci were performed using Haploview v4.2".
67	432	pairwise LD between loci	Comment [A67]: On each individual pair of SNPs within and between loci? Are the results for loci outside the vicinity of the red locus not shown?	In the beginning of the section we now mention that the LD was estimated across the Red locus. LD was measured beyond the range of current definition of the Red locus.
68	441	12 females ($n_{black} = 6$, $n_{red} = 6$)	Comment [A68]: From what source/population?	Throughout the manuscript we use only samples from a single wild population, except for the WGS data. This is mentioned in "DNA samples" section of Methods.
69	455	The assembled sequences	Comment [A69]: For which analyses was this data set used??	This is the main dataset for the measures of diversity across the Red locus and the coalescent simulations. We have rephrased to state this more clearly: "The assembled sequences were aligned using UGENE v1.12.377 and manually edited using BioEdit v7.1.1178. These data were used for the measures of DNA polymorphism across the Red locus and coalescent simulations to test for selection (see below)"
70	465	We obtained the gene expression profiles of 6 birds (3 black and 3 red)	Comment [A70]: Males or females or both?	We collected black and red feathers from 6 birds ($n_{black_male} = 3$, $n_{red_male} = 2$, $n_{red_female} = 1$).
71	476	the Benjamini– Hochberg	Comment [A71]: Reference?	Added "Benjamini, Y. & Hochberg, Y. Controlling the

		method		false discovery rate: A practical and powerful approach to multiple testing. J. R. Stat. Soc. Series B Stat. Methodol. 57 , 289-300 (1995)".
72	491	12 female birds (nblack = 6, nred = 6),	Comment [A72]: From what population? Same birds as above for the inversion test and MiSeq assemblies?	Throughout the manuscript we use only samples from a single wild population, except for the WGS data. This is mentioned in "DNA samples" section of Methods.
73	491	de novo sequencing	Comment [A73]: Isn't de novo sequencing covered above in lines 479-498?	This is for the MiSeq data for the Red locus. We deleted " de novo " and added "using MiSeq" to reduce confusion.
74	492	46,503,706–46,601,744	Comment [A74]: It's not clear how the limits of this region were determined? Why are positions outside of the region of strongly elevated Fst and LD considered part of the Red locus?	Throughout the manuscript the " Red locus" is the full ~100 kbp region sequenced using the MiSeq that contains the ~72 kbp "core region" with high divergence. Because we didn't find any breakpoints nor causal sites that distinguish the core and the flanking regions, the definition of the Red locus is to a degree arbitrary. Because we did not find significant differences in the recombination between entire regions and the core, we use parameters estimated using the entire region in the simulation. We believe this is a more conservative approach than using only the highly divergent region. See main responses.
75	493	One additional sequence stretch (henceforth the 'flanking' region), approximately 7 kb long, was obtained for the same samples adjacent to the Red locus (Supplementary Table 6).	Comment [A75]: What is the value of this "disconnected" sequence on one side of the locus, and again, what is the distinction between the locus and the "core" of the locus?	We agree that including the extra 'flanking region' is confusing, and that this is not used for any analysis beyond as a comparator for population genetic summary statistics. We have therefore removed reference to it, for clarity.
76	496	randomly chose 24 loci	Comment [A76]: How were these sequenced and assembled? Was this a different MiSeq run from	Added "In an additional MiSeq run, "

			above?	
77	502	diversity at the Red locus	Comment [A77]: Using the MiSeq data?	added "using the MiSeq data"
78	508	at the reference and the Red loci	Comment [A78]: This seems problematic. Given recombination and given that the samples were allocated to groups based on genotype at the Red locus, a comparison between the two groups at the reference loci is not necessarily going to measure the same thing (e.g., maximal historical divergence for the Red locus but not for the reference loci). To make the comparison more comparable, why not compare the two maximally divergent alleles at each reference locus?	Please see the response to comment A18 above.
79	511	π_{red} refers to nucleotide diversity within the red allelic class, and π_{total} refers to nucleotide diversity within the black and the red allelic classes combined	Comment [A79]: But w.r.t. the reference loci, this really means between groups of females with different alleles at the Red locus, given that the reference loci are not tightly linked.	We agree and have amended the text as per the reviewer's point: π_{red} refers to nucleotide diversity within alleles of red birds, and π_{total} refers to nucleotide diversity within alleles of both black and red birds combined
80	517	sliding window analyses	Comment [A80]: How did this work for the reference loci, which were relatively short (~1000 bp each)?	We didn't carry out sliding window analyses at the reference loci - we have amended the text to make it clear that we are only referring to analyses at the Red locus at this point.
81	524	genotyped 161 birds	Comment [A81]: Presumably this is the largest sample size available, but 9 μ sat markers is not much data when RADseq and WGS data are also available! For example, I'd like to see PCA, STRUCTURE and/or fineRADstructure results for the n = 32 wild birds with RAD-seq data.	We used microsatellite data for all available samples. We have now added the fineRADstructure analysis using the RADSeq data that included more SNP loci but for a subset of the total sample.

82	533	equivalent	Comment [A82]: Equivalent in what respects?	We have added more detail at this point to explain what we mean by "equivalent to". The new section also includes the suggestion in A83, below.
83	537	sampling scheme	Comment [A83]: Do you mean the sample sizes of males and females and the numbers of each Red locus genotype in the empirical sample?	In our simulations, we used sequence data from females of both morphs. There is a danger that studies on polymorphism fail to correct the bias due to the non-random sampling that otherwise results in association studies based on a focal phenotype. Using population genetic parameters estimated from the non-random sample would violate the assumptions of many common population genetic tests. Here, we provide a way to deal with such biases. We are referring to the sampling scheme of 6 + 6 red and black birds sampled from a wild population where the allele frequencies in nature are 0.144 and 0.856. We have amended the text at this point to make this more explicit, and reordered the methods section so that the part detailing with our consideration of non-random sampling precedes this section.
84	538	population genetic parameters	Comment [A84]: Which parameters? Isn't the point to compare the observed and simulated parameters? Which parameters were kept constant in the simulations and which were tested?	We have modified the text to be more explicit about what parameters we meant - theta and rho. We are interested in testing our null hypothesis of neutral evolution + non-random sampling scheme, but are not carrying out parameter estimation. To this end we determined a value of theta from the reference loci and a value of rho from the R_m test at the Red locus in order to parameterise our simulations. This concern is also responded to in the response to the reviewer's letter.
85	568	for the sequence of the entire core region of the Red locus	Comment [A85]: Why "the entire core region" here when the data were based on the M1-M3 loci? How long are the fragments "amplifiable" using these primer pairs?	We wanted to have the best model for the entire core of the Red locus and assumed the same model can be applied to other parts of the region. This model was used to construct trees in Figure 4, Supplementary Figs. 13~15 which use parts or entire core region.

86	570	For the visualization of the pairwise genetic distance matrix only, we estimated the evolutionary distance between pairs of sequences under the F81 model using phangorn	Comment [A86]: Probably not necessary – doesn't really add much to the phylogenetic tree.	This was for visualization only. Now it is deleted.
87	35	a	an	accepted
88	37	of	that	accepted
89	37	that might	is involved in	accepted
90	38	maintain	maintaining	accepted
91	40	common naturally-occurring	common, naturally occurring	accepted
92	45	will	should	accepted
93	83	and no	. No	accepted
94	83	were found, consistent	were found, but this is consistent	accepted
95	89	refined	measured	accepted
96	92	pair	pairs	accepted
97	112	regulate the feather	regulate feather	accepted
98	113	keratinocyte	keratinocytes	accepted
99	113	that is	that are	accepted
100	116	diverged	divergent	accepted
101	116	developments	development	accepted
102	117	were	was	accepted
103	121	strong functional candidate	strong candidate	accepted
104	122	the development of hair	hair and feather development	accepted

		and feather		
105	133	the spatio-temporal	spatio temporal	accepted
106	138	suggesting the convergent	suggesting convergent	accepted
107	139	and possibly	and a possibly	accepted
108	140	that there may be another selective force mechanism promoteing	that a selection may promote	accepted
109	153	the edges of the core of the Red locus	at the edges of the Red locus	"the edges of the core of the Red locus". We now use " Red locus" and "Core", excluding what we previously referred to as the "flanking region". See response to Comment [A8].
110	157	while the diversity	while diversity	accepted
111	158	alleles were	alleles was	accepted
112	159	The estimate of raw divergence	The estimate of absolute divergence	accepted
113	173	mixed-morph pairings resulted	mixed-morph pairings in captivity resulted	In the beginning of the sentence, we already mentioned that the incompatibilities were observed "in captive population of the Gouldian finch". So adding "in captivity" seems repetitive.
114	174	If this persisted in the wild	If this persisted these results reflect the fitness consequences of interbreeding in the wild	Amended to: "If these incompatibilities persisted in the wild".
115	180	much reduced	much lower	accepted
116	195	neutral processes and	neutral processes or	accepted
117	196	i.e.	i.e.,	accepted
118	201	when the Zr allele was assumed to be derived	when we assumed that the Zr allele was derived	accepted
119	204	contrasting with the neutral	contrasting with neutral	accepted

120	204	expectation assuming a single	expectations for a single	accepted
121	204	causative site that showed a gradual	causative site, which results in a gradual	accepted
122	205	gradual degradation towards	gradual decline towards	accepted
123	216	genealogy at the M1	genealogy of the M1	accepted
124	222	however, the two Gouldian finch	the two Gouldian finch	accepted
125	224	haplotypes have evolved	haplotypes evolved	accepted
126	225	diverged haplotypes	divergent haplotypes	accepted
127	227	diverged M1	divergent M1	accepted
128	227	an insertion	we identified an insertion	accepted
129	228	feature of long terminal	feature of a long terminal	accepted
130	228	zebra finch, was identified (Fig. 2).	zebra finch (Fig. 2).	accepted
131	229	The signal of selection	Signals of selection	accepted
132	229	in terms of the	including	accepted
133	230	neutral simulations,	neutral simulations	accepted
134	230	according to the HKA test	the HKA test	accepted
135	230	was strongest	were strongest	accepted
136	232	colour associated	colour-associated	accepted
137	233	retrotransposon, that	retrotransposon, which	accepted
138	233	of the Gouldian finch from zebra finch	of Gouldian finch and zebra finch	accepted
139	243	origin of the haplotypic	origin of haplotypic	accepted

140	249	diverged haplotypes	divergent haplotypes	accepted
141	253	sites, at the expense	sites, albeit at the expense	accepted
142	253	expense of paternal care	expense of reduced paternal care	accepted
143	254	can quickly rise	would be expected to quickly rise	accepted
144	256	opposing selective forces have	opposing selection has	accepted
145	259	Thus	Thus,	accepted
146	260	density of the red individuals	equilibrium frequency of red individuals	accepted
147	264	where the fitness	in which the fitness	accepted
148	268	generally, and the proportion	generally, the proportion	accepted
149	270	suggesting that selection may have disfavoured	suggesting selection against	accepted
150	276	the species, in the absence	the species and in the absence	accepted
151	277	in respect to	about	accepted
152	284	Identification the Red locus	Identification of the Red locus	accepted
153	289	for the core region	based on two contiguous SNPs in the core	accepted
154	290	Two consecutive SNPs (TG/CA)	These SNPs (TG/CA) were	accepted
155	291	were typed in an	and were typed using	accepted
156	298	association tests	P-values for association tests	"-log ₁₀ P"
157	309	putative LTR	a putative LTR	accepted
158	310	and f,	and h,	accepted
159	326	putative LTR	a putative LTR	accepted

160	342	sampled in 2008	in 2008	accepted
161	357	from a wild population (see above)	from the wild population described above	accepted
162	358	libraries of genomic DNA of 10	DNA fragment libraries for 10	accepted
163	361	a library including	a pooled library including	accepted
164	362	a lane of Illumina HiSeq2000	a lane on an Illumina HiSeq2000 instrument	accepted
165	367	allowing 15%	allowing up to 15%	accepted
166	375	lane of Illumina	lane of an Illumina	accepted
167	376	NextSeq500	NextSeq500 instrument	accepted
168	379	individual reads to the assembly of the	individual reads to the	accepted
169	410	around a RAD tag	around a RADSeq locus	accepted
170	424	the LD and the association	LD across the Red locus and the association	accepted
171	481	for the gene expression	for the expression	accepted
172	491	sequence from the Red locus	sequence encompassing the Red locus	accepted
173	499	10 (5 each of black and red) of which	10 of which (5 each of black and red)	accepted
174	531	widely-used	widely used	accepted
175	629	Poephila gouldiae gould	Poephila gouldiae Gould	accepted

Reviewer #2 (Remarks to the Author): Line numbers are based on " NCOMMS-18-10766-T - 18034016 - Main.pdf".

There continues to be uncertainty and controversy regarding the relative importance of multilocus supergenes (typically associated with genomic regions of low recombination such as inversions) versus pleiotropic single locus “switches” in the evolution of color polymorphisms associated with correlated phenotypic traits. Lately, some supergene examples have been characterized and received a good deal of attention, notably in birds. In this MS Kim and colleagues present evidence for a roughly 72kbp intergenic sequence, that likely plays a role in the regulation of the follistatin gene, as being the only consistent difference genomic between the red and black morphs of Australia’s Gouldian finch, a species in which the morphs differ in morphology, behavior, and physiology as well as head color. This species and its polymorphism are famous and visually striking, but a bit enigmatic in that some of the results for more or less domesticated strains and laboratory populations seem not to hold up well in studies of wild birds. The authors additionally find no evidence of an inversion being associated with the region of high differentiation and the evidence against an inversion seems to me persuasive.

I can see no obvious or severe problems with the claims presented by the authors, and the MS is generally well written with few errors.

We thank the reviewer for their kind and supportive comments. Below we address specific points.

One point that seems to me unclear is why balancing selection and an inversion are treated as strict alternatives. If a population is polymorphic for the presence or absence of an inversion, one must still explain the persistence of that polymorphism, and balancing selection could be the answer. Similarly, inversions may not be a perfect barrier to gene flow between homologous inverted and non-inverted regions of chromosomes. What they are really seeking to explain is strong linkage disequilibrium along a substantial length of chromosome, with the alternative forms associated with different complex phenotypes.

We agree with the suggestion that an inversion and balancing selection should not be presented as alternatives, and have amended the abstract accordingly.

The main criticism one can make is that the story is somewhat incomplete. The authors were unable to characterize transcriptomic differences between the morphs that relate directly to the putative switch

locus, thus the mode of action of the apparent switch has not been elucidated, and they have not attempted a manipulation of the locus in question, so its role is not fully established. The main question for the Editor then is whether the story as presented—which is clearly important and interesting—is of sufficiently broad interest and impact as to justify publication in Nature Communications.

We agree that it would be fascinating and important to explore the molecular function of the candidate gene we found, but also that it is beyond the scope of this study. The transcriptomic analysis was not conclusive, but even if such data were available they would only take us partially towards understanding causality. Manipulating genes in this or other non-model avian taxa is not straightforward, and would need to be the subject of a new project.

We have made significant progress in identifying the locus and in understanding its maintenance, adding very substantially to a body of earlier high-profile work on this system. Our manuscript is the first to show the clear association of the focal region with colour polymorphism of functional importance in this species. This locus has a potentially important role in speciation, as some speciation events are associated with differentiation between species in this region. It will be intriguing to study functional roles of the genes associated with this regulatory network in relation to the formation of incompatibilities, and we believe our paper provides an important starting point towards this end.

Specific Comments:

39: comma after common

Accepted

39: claims to be models or emerging models are regrettably common and often inadequately substantiated; better to make a more specific claim for importance or novelty

Rephrased as “This study demonstrates that, although this region is involved in the maintenance of a species boundary in other hybridising taxa, variation at the same locus within a species can lead to a balanced polymorphism associated with complex morphological, behavioural and physiological effects.”

46: delete “to”

Accepted

84: consistent seems strong with $p=.11$... but not significantly different from expectation is reasonable

This is now rephrased as “The low frequency of the homozygous red male is not significantly different from expectation for a sex-linked locus under random mating (Supplementary Table 3) and a prediction based on the phenotypic frequency”.

188: good to provide a citation here

We are not aware of any explicit treatment of this problem in the literature, although it has been previously recognised. For example, Hudson et al. (1994) and Saunders et al. (2005) both reconstructed random samples (that matched allele frequencies in nature) by subsampling from non-random data, before making population genetic inferences. Consequently, it is hard to know what to cite. Initially, our consideration of this problem was based on intuition, and was borne out by the simulations (e.g. the results presented in Figure 3a-d.). So, we have chosen to refer to the simulations presented in Figure 3 at this point.

** Hudson, R. R., Bailey, K., Skarecky, D., Kwiatowski, J., Ayala F. J. Evidence for positive selection in the superoxide dismutase (Sod) region of *Drosophila melanogaster*. *Genetics* **136**, 1329–1340 (1994).*

** Saunders, M. A., Slatkin, M., Garner, C., Hammer, M. F., Nachman, M. W. The extent of linkage disequilibrium caused by selection on G6PD in humans. *Genetics* **171**, 1219–1229 (2005).*

254: better to explain a little more clearly how this scenario leads to frequency-dependence

It is now clarified that the high frequency of red males creates a competitive and stressful environment and that red males suffer more from those conditions.

263: run on sentence

We have split the sentence into two.

267: sentence runs on some

We have split the sentence into two.

280: family-wise?

Family-wise error rate refers to the error rate for the entire collection of family of hypotheses and we used a permutation test to control it using the `-mperm` option in `plink` (Clarke et al 2011). The text is now rephrased to “A circle around a dot indicates the only significant signal in GWAS”.

** Clarke, G. M. et al. Basic statistical analysis in genetic case-control studies. Nat. Protoc. 6, 121-133 (2011).*

293: a little more explanation of how to interpret c would be helpful

We have added a bit more explanation including the definition of the Red locus and the type of association test for clarity.

347: nice if this point could be fit into the main text, though clearly space is an issue

We have added a sentence about an epistatic interaction between the Red locus and an autosomal locus in the introduction.

Suppl 301: not Kim at al

Deleted “Kim at al.”

Suppl 448: why non-alphabetical legend organization? Correct?

The order of the legend has been corrected.

Suppl 458: 10a?

It is now Supplementary Fig. 11a (previously Supplementary 10a) that shows a uniform distribution.

Suppl 506: could mention lack of field evidence for assortative mating here

We have now clarified that these incompatibilities were not fully tested in a wild population in the introduction.

Reviewer #3 (Remarks to the Author):

This is an impressive and well carried out study examining the genetic basis of a color polymorphism in Gouldian finches using a variety of approaches. The authors demonstrate that the divergence between the red and the black morph reside to a 72 kbp region on the Z chromosome, consistent with earlier findings of this polymorphism being located in this region on Z. There are several interesting findings: firstly, given the many differences in a plethora of traits between the morphs, that this is not due to an inversion as seen in many other species but rather (most likely) due to pleiotropic effect. Second, that it locates to a intergenic region suggesting a regulatory role (also well worth keeping in mind that 60-70% of all GWAS hits are intergenic) rather than a coding sequence change. Third, using coalescent simulations, the authors demonstrate that this polymorphism is likely maintained by balancing selection and arose in the Gouldian finch lineage.

Overall, this ms is very well written and carried out, although I note that I do not have expertise in all the analyses performed by the authors. It provides a nice contrast to the recent findings of the involvement of inversions ('supergenes') causing polymorphisms in several bird species (ruffs, zebra finches etc).

We thank the reviewer for their generous and enthusiastic comments. The suggestions on how to improve the manuscript are helpful, and we have taken these on board.

From my reading the mapping and identification of the region seems very solid and it is very interesting that the same gene seem to be involved in color divergence in a different species pair (golden and blue winged warblers). Given the finding of a TE within the Red locus I did wonder if this could be linked to the polymorphism as well, even if it arose before divergence between the red and black morph it could facilitate divergence. This is interesting since TE is linked to color patterning and polymorphism in both *Heliconius* and *Biston betularia*.

We agree that there is a possibility that the inserted TE might have had a role in the polymorphism given the highest divergence between morphs in this region. However, refining the Red locus exclusively within the TE region by using historical recombination is impractical because the high level of LD in the Red locus means that we would need hundreds or thousands of wild birds to obtain enough recombinants across this region. The required sample size is very difficult to achieve in the Gouldian finch because this species is currently recovering from endangered status and there are only ~2,500 individuals in the wild population. Biochemical

assay may be required to identify the causal site(s), which is beyond the scope of this study. We do find the reference the reviewer suggests to be very useful and cite it in the main text.

It is also not clear how This is even more so given that the analysis to demonstrate balancing selection as the mechanism for the maintenance of this polymorphism require strong assortative mating, something which is not observed in natural populations.

Kokko et al. (2014) showed such results using a simulation with all parameters derived from a captive population where the incompatibilities were observed. Therefore, it may not be appropriate to apply the results directly to the wild population. We think this may cause confusion so have decided to remove it from the text.

** Kokko, H., Griffith, S. C. & Pryke, S. R. The hawk–dove game in a sexually reproducing species explains a colourful polymorphism of an endangered bird. Proc. R. Soc. B 281, 20141794 (2014).*

Should one not also expect accumulation of deleterious mutations within the region since recombination is essentially not taking place? This is something that you could test with the resequencing data?

The Red locus is intergenic, so it is difficult to define which sites we expect to be under purifying selection. It might be possible to do so using the PhastCons information that we use elsewhere, but we still lack clear-cut information about the functional importance of individual sites. There are further complications about using our resequencing data to answer this question. First, we suspect that our sample size is not large enough to detect differences in the efficacy of purifying selection at the Red locus vs. other loci (e.g. the Z-linked reference introns) using, for example, differences in derived allele frequencies. Second, it is not clear what signal of relaxed purifying selection is expected at the Red locus, given the linkage to a site or sites under balancing selection. Overall, although this question is interesting, we do not think that we are best-placed to answer it with the data presented here, and doing it justice would likely involve a large amount of extra effort, which we feel is outside the scope of the current study.

We look forward to learning the editorial decision of the manuscript.

Yours sincerely

Terry Burke, Kang-Wook Kim, Benjamin C. Jackson, Hanyuan Zhang, David P. L. Toews, Scott A.

Taylor, Emma I. Greig, Irby J. Lovette, Mengning M. Liu, Angus Davison, Simon C. Griffith & Kai Zeng

Reviewers' Comments:

Reviewer #1:

Remarks to the Author:

I've read the authors' responses to all three reviewer's comments and generally find the responses to be thoughtful and appropriate.

On one issue that I considered to be significant (defining the limits of the "red locus"), the authors offered some additional explanation and clarification but avoided more substantial revision. Although the authors argue that an arbitrary threshold is needed to specify where the region of elevated F_{st} and LD starts and stops, I find the definition of the red locus used in the paper substantially more arbitrary, with no biologically relevant rationale for the location of its start and end points.

The authors also make the argument that by simulating a larger region, their test of the null hypothesis of neutrality becomes more "conservative" (i.e., less likely to reject the null). By this logic, the test could be made even more conservative by including even more flanking sequence, but making tests more conservative is not desirable if it also makes them less biologically relevant. I see potential downsides of not defining the red locus more narrowly: 1) regions with clearly heterogenous characteristics (and history) are compared to a simulation that assumes uniform processes across the entire region; and 2) the simulation analysis and its results becomes more complicated to present and more difficult for the reader to understand and interpret. It certainly raised a number of questions for me.

While I continue to think that defining the "red locus" as the region of clearly elevated F_{st} and LD is the only sensible approach, I understand that adopting my suggestion would require repeating a number of analyses. On the other hand, by arbitrarily defining the red locus as encompassing a larger region than it really does, the authors add unnecessary confusion and complication to what is a striking and obvious result. I note that in the abstract the authors refer to the smaller ~ 72 kb region of the Z chromosome as being associated with the color polymorphism. Likewise, Toomey et al. (2018) identify the same ~ 70 kb region as the "candidate locus." Either 70 or 72 kb reflects a much more sensible definition than the ~ 100 kb adopted for the simulation analysis.

While I continue to disagree with the authors' decisions and rationale on this issue, the additional information and clarifications the authors provide are probably sufficient to make the analyses understandable for most readers.

The only other concern I would raise is that a very similar paper was recently published. Toomey et al. (2018) present similar analyses for the same species, including RNA-seq data and a genome scan based on whole genome re-sequencing data, and obtain broadly similar results. The Toomey et al. paper is based on samples obtained from captive birds in Portugal rather than a wild population, but in most other respects, it is in my opinion a better paper, with more clearly presented analyses that are based on more robust data sets. (Please note that I have no connection to the Toomey et al. study, do not know any of its authors, and just recently learned about it.)

Given that the Toomey et al. paper has been published, the authors of this paper should of course cite it, but also explicitly address the question of how the analyses they present extend our understanding beyond the results of Toomey et al. For example, the simulation analysis is one component that is unique as compared to Toomey et al.

Toomey MB, Marques CI, Andrade P et al. (2018) A non-coding region near Follistatin controls head colour polymorphism in the Gouldian finch. *Proceedings of the Royal Society B-Biological Sciences*

285, 20181788.

Reviewer #2:

Remarks to the Author:

I have reviewed the revisions the authors made in response to my previous comments, and am satisfied with them.

Reviewer #3:

Remarks to the Author:

This revised version has improved the clarity of the text further and I think this will make a very interesting contribution to the field of genetics of polymorphisms in general, and of course color polymorphisms in particular.

Reviewers' comments:

Reviewer #1 (Remarks to the Author):

I've read the authors' responses to all three reviewer's comments and generally find the responses to be thoughtful and appropriate.

On one issue that I considered to be significant (defining the limits of the "red locus"), the authors offered some additional explanation and clarification but avoided more substantial revision. Although the authors argue that an arbitrary threshold is needed to specify where the region of elevated F_{st} and LD starts and stops, I find the definition of the red locus used in the paper substantially more arbitrary, with no biologically relevant rationale for the location of its start and end points.

We are willing to adopt the reviewer's suggested definition of the *Red* locus – i.e. the region that we previously described as the “core” – and can see that this has the virtue of emphasising that this is where the causal variants are most probably located. We have revised the paper accordingly (Figures 1-2; estimates in the text on pp. 6-7 taken from Fig. 2; previous references to “core” region amended to refer to *Red* locus; where appropriate the 100-kbp sequence is now referred to as the “candidate region”; the shading to indicate the *Red* locus location was adjusted in Supplementary Tables 4-7; the values in Supp. Table 9 have been recalculated for the smaller region).

We emphasise that this change of definition does not affect our analysis. This is because we are simply testing whether patterns of genetic variation in the 100-kbp region (the red locus and its flanks) are consistent with neutral evolution (while recognising that variants underlying the trait are in this region, and that our sampling is nonrandom). We will elaborate further on this point in our response to the next comment.

The authors also make the argument that by simulating a larger region, their test of the null hypothesis of neutrality becomes more "conservative" (i.e., less likely to reject the null). By this logic, the test could be made even more conservative by including even more flanking sequence, but making tests more conservative is not desirable if it also makes them less biologically relevant. I see potential downsides of not defining the red locus more narrowly: 1) regions with clearly heterogenous characteristics (and history) are compared to a simulation that assumes uniform processes across the entire region; and 2) the simulation analysis and its results becomes more complicated to present and more difficult for the reader to understand and interpret. It certainly raised a number of questions for me.

While I continue to think that defining the "red locus" as the region of clearly elevated F_{st} and LD is the only sensible approach, I understand that adopting my suggestion would require repeating a number of analyses. On the other hand, by arbitrarily defining the red

locus as encompassing a larger region than it really does, the authors add unnecessary confusion and complication to what is a striking and obvious result.

By “conservative”, we simply meant that, by including some data from a larger genomic region, a lower proportion of windows in our sliding-window analysis are likely to show a significant difference from neutral expectation (see Figure 3). The inclusion of the flanking regions also makes the summary measures of differentiation for the region (π_{total} , $d_{\text{xy Red locus}}$, $D_{\text{Taj total}}$, F_{ST} ; pp. 6-7) more similar to those for the reference loci, which is again conservative.

The reviewer is correct in asserting that it would be undesirable to carry out the analysis on data from a very large genomic region but the extent of the flanking regions on either side of the *Red* locus is, in fact, rather modest. We must emphasise that having data from the flanking regions is a critical part of our analysis and presentation, for the following reasons:

1. As shown clearly in Figure 3, the evidence for selection on variants within the red locus comes from the observed patterns of variability (black lines) lying well outside what is expected under neutrality (grey area). If we restricted the simulation to the 72-kbp *Red* locus then these patterns would not be apparent, and the important Fig. 3 would be much less informative.
2. In this context, the fact that the patterns of variability in the flanking regions fall well within what is expected under neutrality highlights that what we see at the *Red* locus falls outside of neutral expectation, and also adds credibility to our simulation model.
3. The presence of these flanks also enables the reader to visually appreciate the extent of the *Red* locus – indeed without them we believe that the figures would be *more* difficult to interpret – and avoids the need for us to make a slightly arbitrary decision about the precise extent of the *Red* locus.

We have now included this reasoning in the Supplementary Text: Estimating ρ , the population recombination rate (p. 75). As mentioned in our response to the previous comment, the null hypothesis in our simulation-based analysis (Figure 3) does not depend on how the *Red* locus is defined. We are just asking whether this 100-kbp region is neutral, knowing that the variants underlying the trait are located within it, and that our sampling is conditional on the phenotypes of the birds (6 red and 6 black).

In response to the concern about heterogeneity, the most important parameter here is the recombination rate, because it determines how quickly LD decays in our simulated data as we move away from the site assumed to be responsible for the trait (see the decay in the grey area in Figure 3d). However, as mentioned in our response to the comments on the previous version of the paper, there was no evidence that this parameter is different between *Red* and its flanking regions (indeed, it is numerically

very close: goodness of fit $\chi^2_1 = 0.05$, $P = 0.82$; Supplementary Table 13; as now described in Supplementary Text: Estimating ρ , the population recombination rate, p. 76).

As an additional measure to address the Reviewer's concern, we have now explained our reasoning more carefully in the Methods (Testing for Selection, pp. 24-25), where we also demonstrate that the results are not affected by the recombination rate. These changes involve the addition of Supplementary Figs. 11-12 and associated text to show how robust the analysis is to the estimation of the population recombination rate, ρ . When we used $\rho = 0.0002$, which is 20 times lower than our estimate from the 100-kbp-sequence data, the results did not change. It seems obvious (from Supplementary Tables 12-13) that the population recombination rate ρ for the *Red* locus only is $\gg 0.0002$. The rationale for using the value of 0.0002 is now explained in the same paragraph.

Finally, to further allay the Reviewer's concern, we have repeated the simulations for the 72-kbp *Red* locus alone (in the new Supplementary Figs. 13-14, as now referred to in the Testing for Selection section on p. 25). The estimate of ρ for this reduced region was unchanged from the value for the entire sequenced region. (Note also that some Supplementary Figures have been renumbered to accommodate these additions).

I note that in the abstract the authors refer to the smaller ~72kb region of the Z chromosome as being associated with the color polymorphism. Likewise, Toomey et al. (2018) identify the same ~70kb region as the "candidate locus." Either 70 or 72 kb reflects a much more sensible definition than the ~100kb adopted for the simulation analysis.

While I continue to disagree with the authors' decisions and rationale on this issue, the additional information and clarifications the authors provide are probably sufficient to make the analyses understandable for most readers.

As described above, we have accepted the Reviewer's suggestion to redefine the locus (as what we previously described as the 72-kbp "core").

The only other concern I would raise is that a very similar paper was recently published. Toomey et al. (2018) present similar analyses for the same species, including RNA-seq data and a genome scan based on whole genome re-sequencing data, and obtain broadly similar results. The Toomey et al. paper is based on samples obtained from captive birds in Portugal rather than a wild population, but in most other respects, it is in my opinion a better paper, with more clearly presented analyses that are based on more robust data sets. (Please note that I have no connection to the Toomey et al. study, do not know any of its authors, and just recently learned about it.)

Given that the Toomey et al. paper has been published, the authors of this paper should of course cite it, but also explicitly address the question of how the analyses they present extend

our understanding beyond the results of Toomey et al. For example, the simulation analysis is one component that is unique as compared to Toomey et al.

Toomey MB, Marques CI, Andrade P et al. (2018) A non-coding region near Follistatin controls head colour polymorphism in the Gouldian finch. Proceedings of the Royal Society B-Biological Sciences 285, 20181788.

During the revision of the current manuscript we were alerted to a study on face colour polymorphisms in domesticated Gouldian finches by Toomey et al. that was published in October 2018. We were unaware of Toomey et al.'s work and, indeed, in the light of his interest in avian pigmentation we had suggested Dr. Toomey as a possible referee.

The research presented here confirms, complements, and extends Toomey et al. in several ways. First, both independent approaches highlight the same genomic region upstream of the *follistatin* gene, providing strong evidence of its functional role in generating the colour polymorphism. Second, unlike Toomey et al., our analysis incorporates data from wild birds, making clear the relevance of this region to non-domesticated populations. Finally, our simulations and evolutionary reconstructions add key historical context to how and why this polymorphism has evolved and is maintained in the wild. We have added a paragraph to the text on page 11 where we now refer to Toomey et al.

Reviewer #2 (Remarks to the Author):

I have reviewed the revisions the authors made in response to my previous comments, and am satisfied with them.

No response required.

Reviewer #3 (Remarks to the Author):

This revised version has improved the clarity of the text further and I think this will make a very interesting contribution to the field of genetics of polymorphisms in general, and of course color polymorphisms in particular.

No response required.

Reviewers' Comments:

Reviewer #1:

Remarks to the Author:

I have reviewed the authors' responses to my previous round of comments and am pleased to see modifications made to the paper that I think will improve the clarity of the presentation. Likewise, I like the direct approach of citing the recent Toomey et al. study. I have no further comments.